# Label-Aware Neural Tangent Kernel: Toward Better Generalization and Local Elasticity

**Shuxiao Chen**      **Hangfeng He**      **Weijie J. Su**

University of Pennsylvania

{shuxiaoc@wharton, hangfeng@seas, suw@wharton}.upenn.edu

## Abstract

As a popular approach to modeling the dynamics of training overparametrized neural networks (NNs), the neural tangent kernels (NTK) are known to fall behind real-world NNs in generalization ability. This performance gap is in part due to the *label agnostic* nature of the NTK, which renders the resulting kernel not as *locally elastic* as NNs (He & Su, 2020). In this paper, we introduce a novel approach from the perspective of *label-awareness* to reduce this gap for the NTK. Specifically, we propose two label-aware kernels that are each a superimposition of a label-agnostic part and a hierarchy of label-aware parts with increasing complexity of label dependence, using the Hoeffding decomposition. Through both theoretical and empirical evidence, we show that the models trained with the proposed kernels better simulate NNs in terms of generalization ability and local elasticity.[1]

## 1   Introduction

The last decade has witnessed the huge success of deep neural networks (NNs) in various machine learning tasks (LeCun et al., 2015). Contrary to its empirical success, however, the theoretical understanding of real-world NNs is still far from complete, hindering its applicability to many domains where interpretability is of great importance, such as autonomous driving and biological research (Doshi-Velez & Kim, 2017).

More recently, a venerable line of work relates overparametrized NNs to kernel regression from the perspective of their training dynamics, providing positive evidence towards understanding the optimization and generalization of NNs (Jacot et al., 2018; Chizat & Bach, 2018; Cao & Gu, 2019; Lee et al., 2019; Arora et al., 2019a; Chizat et al., 2019; Du et al., 2019; Li et al., 2019; Zou et al., 2020). To briefly introduce this approach, let $\boldsymbol{x}_i, y_i$ be the feature and label, respectively, of the $i$th data point, and consider the problem of minimizing the squared loss $\frac{1}{n}\sum_{i=1}^{n}(y_i - f(\boldsymbol{x}_i, \boldsymbol{w}))^2$ using gradient descent, where $f(\boldsymbol{x}, \boldsymbol{w})$ denotes the prediction of NNs and $\boldsymbol{w}$ are the weights. Starting from an random initialization, researchers demonstrate that the evolution of NNs in terms of predictions can be well captured by the following kernel gradient descent

$$\frac{d}{dt}f(\boldsymbol{x}, \boldsymbol{w}_t) = -\frac{1}{n}\sum_{i=1}^{n}K(\boldsymbol{x}, \boldsymbol{x}_i)\big(f(\boldsymbol{x}_i, \boldsymbol{w}_t) - y_i\big) \tag{1}$$

in the infinite width limit, where $\boldsymbol{w}_t$ is the weights at time $t$. Above, $K(\cdot, \cdot)$, referred to as *neural tangent kernel* (NTK), is time-independent and associated with the architecture of the NNs. As a profound implication, an infinitely wide NNs is "equivalent" to kernel regression with a deterministic kernel in the training process.

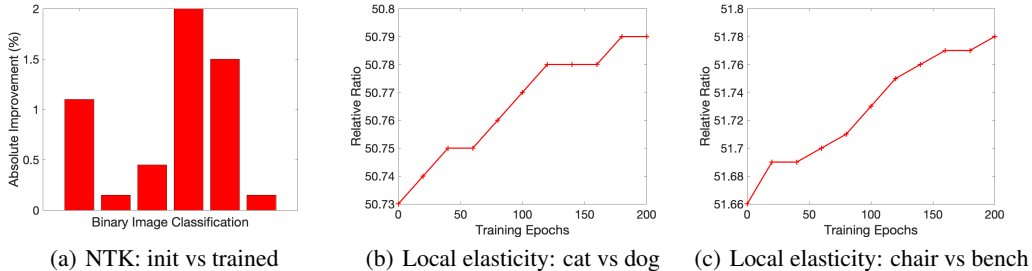

(a) NTK: init vs trained     (b) Local elasticity: cat vs dog     (c) Local elasticity: chair vs bench

Figure 1: The impact of labels on a 2-layer NN in the aspect of generalization ability and local elasticity. Fig. (a) displays the absolute improvement in six binary classification tasks by using $K_t^{(2)}$ instead of $K_0^{(2)}$, indicating that NNs learn better feature representations with the help of labels along the training process (see Sec. 3.2 for more details). Fig. (b) and (c) show that in two label systems (cat v.s. dog and bench v.s. chair), the strength of local elasticity (measured by the relative ratio defined in Eq. (9)) both increases with training, but in different patterns, indicating that the labels play an important role in the local elasticity of NNs (See Appx. D.1 for more details).

Despite its immense popularity, many questions still remain unanswered concerning the NTK approach, with the most crucial one, perhaps, being the non-negligible performance gap between a kernel regression using the NTK and a real-world NNs. Indeed, according to an experiment done by Arora et al. (2019a), on the CIFAR-10 dataset (Krizhevsky, 2009), a vanilla 21-layer CNN can achieve 75.57% accuracy, whereas the corresponding convolutional NTK (CNTK) can only attain 64.09% accuracy. This significant performance gap is widely recognized to be attributed to the *finiteness* of widths in real-world NNs. This perspective has been investigated in many papers (Hanin & Nica, 2019; Huang & Yau, 2019; Bai & Lee, 2019; Bai et al., 2020), where various forms of "finite-width corrections" have been proposed. However, the computation of these corrections all involve incremental training. This fact is in sharp contrast to kernel methods, whose computation is "one-shot" via solving a simple linear system.

In this paper, we develop a new approach toward closing the gap by recognizing a recently introduced phenomenon termed *local elasticity* in training neural networks (He & Su, 2020). Roughly speaking, neural networks are observed to be locally elastic in the sense that if the prediction at a feature vector $x$ is not significantly perturbed, after the classifier is updated via stochastic gradient descent at a (labeled) feature vector $x$ that is dissimilar to $x'$ in a certain sense. This phenomenon implies that the interaction between two examples is heavily contingent upon whether their labels are the same or not. Unfortunately, the NTK construction is clearly independent of the labels, thereby being label-agnostic, meaning that it is independent of the labels $y$ in the training data. This ignorance of the label information can cause huge problems in practice, especially when the semantic meanings of the features crucially depends on the *label system*.

To shed light on the vital importance of the label information, consider a collection of natural images, where in each image, there are two objects: one object is either a cat or dog, and another is either a bench or chair. Take an image $x$ that contains dog+chair, and another $x'$ that contains dog+bench. Then for NTK to work well on both label systems, we would need $\mathbb{E}_{\text{init}} K(x, x') \approx 1$ if the task is dog v.s. cat, and $\mathbb{E}_{\text{init}} K(x, x') \approx -1$ if the task is bench v.s. chair, a fundamental contradiction that cannot be resolved by the label-agnostic NTK (see also Claim 1.1). In contrast, NNs can do equally well in both label systems, a favorable property which can be termed adaptivity to label systems. To understand what is responsible for this desired adaptivity, note that (1) suggests that NNs can be thought of as a time-varying kernel $K_t^{(2)}$. This "dynamic" kernel differs from the NTK in that it is *label-aware*, because it depends on the trained parameter $w_t$ which further depends on $y$. As shown in Fig. 1, such an awareness in labels play an import role in the generalization ability and local elasticity of NNs.

Thus, in the search of "NN-simulating" kernels, it suffices to limit our focus on the class of label-aware kernels. In this paper, we propose two label-aware NTKs (LANTKs). The first one is based on a notion of *higher-order regression*, which extracts higher-order information from the labels by estimating whether two examples are from the same class or not. The second one is based on approximately solving neural tangent hierarchy (NTH), an infinite hierarchy of ordinary different

equations that give a precise description of the training dynamics (1) (Huang & Yau, 2019), and we show this kernel approximates $K_t^{(2)}$ strictly better than $\mathbb{E}_{\text{init}} K_0^{(2)}$ does (Theorem 2.1).

Although the two LANTKs stems from very different intuitions, their analytical formulas are perhaps surprisingly similar: they are both quadratic functions of the label vector $y$. This brings a natural question: *is there any intrinsic relationship between the two?* And more generally, *what does a generic label-aware kernel, $K(\boldsymbol{x}, \boldsymbol{x}', S)$, which may have arbitrary dependence structure on the training data $S = \{(\boldsymbol{x}_i, y_i) : i \in [n]\}$, look like?* Using *Hoeffding Decomposition* (Hoeffding et al., 1948), we are able to obtain a structural lemma (Lemma 2.2), which asserts that any label-aware kernel can be decomposed into the superimposition of a label-agnostic part and a *hierarchy* of label-aware parts with increasing complexity of label dependence. And the two LANTKs we developed can, in a certain sense, be regarded as the *truncated versions* of this hierarchy.

We conduct comprehensive experiments to confirm that our proposed LANTKs can indeed better simulate the quantitative and qualitative behaviors of NNs compared to their label-agnostic counterpart. On the one hand, they generalize better: the two LANTKs achieve a $1.04\%$ and $1.94\%$ absolute improvement in test accuracy ($7.4\%$ and $4.1\%$ relative error reductions) in binary and multi-class classification tasks on CIFAR-10, respectively. On the other hand, they are more "locally elastic": the relative ratio of the kernelized similarity between intra-class examples and inter-class examples increases, a typical trend observed along the training trajectory of NNs.

## 1.1 Related Work

**Kernels and NNs.** Starting from Neal (1996), a line of work considers infinitely wide NNs whose parameters are chosen randomly and only the last layer is optimized (Williams, 1997; Le Roux & Bengio, 2007; Hazan & Jaakkola, 2015; Lee et al., 2017; Matthews et al., 2018; Novak et al., 2018; Garriga-Alonso et al., 2018; Yang, 2019). When the loss is the least squares loss, this gives rise to a class of interesting kernels different from the NTK. On the other hand, if all layers are trained by gradient descent, infinitely wide NNs give rise to the NTK (Jacot et al., 2018; Chizat & Bach, 2018; Lee et al., 2019; Arora et al., 2019a; Chizat et al., 2019; Du et al., 2019; Li et al., 2019), and the NTK also appears implicitly in many works when studying the optimization trajectories of NN training (Li & Liang, 2018; Allen-Zhu et al., 2018; Du et al., 2018; Ji & Telgarsky, 2019; Chen et al., 2019).

**Limitations of the NTK and corrections.** Arora et al. (2019b) demonstrates that in many small datasets, models trained by the NTK can outperform its corresponding NN. But for moderately large scale tasks and for practical architectures, the performance gap between the two is empirically observed in many places and further confirmed by a series of theoretical works (Chizat et al., 2019; Ghorbani et al., 2019b; Yehudai & Shamir, 2019; Bietti & Mairal, 2019; Ghorbani et al., 2019a; Wei et al., 2019; Allen-Zhu & Li, 2019). This observation motivates various attempts to mitigate the gap, such as incorporating pooling layers and data augmentation into the NTK (Li et al., 2019), deriving higher-order expansions around the initialization (Bai & Lee, 2019; Bai et al., 2020), doing finite-width correction (Hanin & Nica, 2019), injecting noise to gradient descent (Chen et al., 2020), and most related to our work, the NTH (Huang & Yau, 2019).

**Label-aware kernels.** The idea of incorporating label information into the kernel is not entirely new. For example, Cristianini et al. (2002) proposes an alignment measure between two kernels, and argues that aligning a label-agnostic kernel to the "optimal kernel" $yy^\top$ can lead to favorable generalization guarantees. This idea is extensively exploited in literature on kernel learning (see, e.g., Lanckriet et al. 2004; Cortes et al. 2012; Gönen & Alpaydin 2011). In another related direction, Tishby et al. (2000) proposes the information bottleneck principle, which roughly says that an optimal feature map should simultaneously minimize its mutual information (MI) with the feature distribution and maximize its MI with the label distribution, thus incorporating the label information (see also Tishby & Zaslavsky 2015 in the context of deep learning). We refer the readers to Appx. C for a detailed discussion of the connections and differences of our proposals to these two lines of research.

## 1.2 Preliminaries

We focus on binary classification tasks for ease of exposition, but our results can be easily extend to multi-class classification tasks. Suppose we have i.i.d. data $S = \{(\boldsymbol{x}_i, y_i) : i \in [n]\} \subseteq \mathbb{R}^d \times \{\pm 1\}$. Let $\mathbf{X} \in \mathbb{R}^{n \times d}$ be the feature matrix and let $y \in \{\pm 1\}^n$ be the vector of labels. Let $\boldsymbol{w} = \{(\mathbf{W}^{(\ell)}, \mathbf{b}^{(\ell)}) \in \mathbb{R}^{p_\ell \times q_\ell} \times \mathbb{R}^{q_\ell} : \ell \in [L]\}$ be the collection of weights and biases at each layer.

A neural network function is recursively defined as $\boldsymbol{x}^{(0)} = \boldsymbol{x}, \boldsymbol{x}^{(\ell)} = \frac{1}{\sqrt{q_\ell}}\sigma(\mathbf{W}^{(\ell)}\boldsymbol{x}^{(\ell-1)} + \mathbf{b}^{(\ell)})$ and $f(\boldsymbol{x}, \boldsymbol{w}) = \boldsymbol{x}^{(L)}$, where $\sigma$ is the activation function which applies element-wise to matrices. Note that $q_1 = d$ and $p_L = 1$.

Consider training NNs with least squares loss: $L(\boldsymbol{w}) = \frac{1}{2n}\sum_{i=1}^n \left(f(\boldsymbol{x}_i, \boldsymbol{w}) - y_i\right)^2$. The gradient flow dynamics (i.e., gradient descent with infinitesimal step sizes) is given by $\dot{\boldsymbol{w}}_t = -\frac{\partial}{\partial \boldsymbol{w}}L(\boldsymbol{w}_t)$, where we use the dot notation to denote the derivative w.r.t. time, and $\boldsymbol{w}_t$ is the parameter value at time $t$. The dynamics w.r.t. $f$ is given by a kernel gradient descent (1), where the corresponding *second-order* kernel is given by $K_t^{(2)}(\boldsymbol{x}, \boldsymbol{x}') = \langle \frac{\partial}{\partial \boldsymbol{w}}f(\boldsymbol{x}, \boldsymbol{w}_t), \frac{\partial}{\partial \boldsymbol{w}}f(\boldsymbol{x}', \boldsymbol{w}_t)\rangle$. With sufficient over-parameterization, one can show that $K_t^{(2)} \approx K_0^{(2)}$, and with Gaussian initialization, one can show that $K_0^{(2)}$ concentrates around $\mathbb{E}_{\text{init}}K_0^{(2)}$, where $\mathbb{E}_{\text{init}}$ is the expectation operator w.r.t. the random initialization. Hence, $K_t^{(2)}$ can be approximated by the *deterministic kernel* $\mathbb{E}_{\text{init}}K_0^{(2)}$, the NTK.

We have heuristically argued in Section 1 that the label-agnostic property of NTK can cause problems when the semantic meaning of the features is highly dependent on the label system. To further illustrate this point, in Appx. A.1, we construct a simple example that validates the following claim:

**Claim 1.1** (Curse of Label-Agnosticism). *If a kernel $K$ is label-agnostic and it works well on one label system, then there exists a natural relabeling, which gives another label system on which $K$ performs arbitrarily bad. In other words, $K$ is not adaptive to different label systems.*

We make two remarks. The above claim applies to any label-agnostic kernels, and the NTK is only a specific example. Moreover, the above claim only applies to the generalization error. A label-agnostic kernel can always achieve zero training error, as long as the corresponding kernel matrix is invertible.

## 2 Construction of Label-Aware NTKs

We now propose two ways to incorporate label awareness into NTKs. In Section 2.3, we give a unified interpretation of the two approaches using the Hoeffding decomposition.

### 2.1 Label-Aware NTK via Higher-Order Regression

We shall all agree that a "good" feature map should map the feature vector $\boldsymbol{x}$ to $\phi(\boldsymbol{x})$ s.t. a simple linear fit in the $\phi(\cdot)$-space can give satisfactory performance. In this sense, the "optimal" feature map is obviously its corresponding label: $\phi(\boldsymbol{x}) := \text{label of } \boldsymbol{x}$. Hence, for $\alpha, \beta \in [n]$, the corresponding "optimal" kernel is given by $K^\star(\boldsymbol{x}_\alpha, \boldsymbol{x}_\beta) = y_\alpha y_\beta$. This motivates us to consider interpolating the NTK with the optimal kernel: $\mathbb{E}_{\text{init}}K_0^{(2)}(\boldsymbol{x}_\alpha, \boldsymbol{x}_\beta) + \lambda K^\star(\boldsymbol{x}_\alpha, \boldsymbol{x}_\beta) = \mathbb{E}_{\text{init}}K_0^{(2)}(\boldsymbol{x}_\alpha, \boldsymbol{x}_\beta) + \lambda y_\alpha y_\beta$, where $\lambda \geq 0$ controls the strength of the label information. However, this interpolated kernel cannot be calculated on the test data. A natural solution is to estimate $y_\alpha y_\beta$. Thus, we propose to use the following kernel, which we term as LANTK-HR:

$$K^{(\text{HR})}(\boldsymbol{x}, \boldsymbol{x}') := \mathbb{E}_{\text{init}}K_0^{(2)}(\boldsymbol{x}, \boldsymbol{x}') + \lambda \mathcal{Z}(\boldsymbol{x}, \boldsymbol{x}', S), \tag{2}$$

where $\mathcal{Z}(\boldsymbol{x}, \boldsymbol{x}', S)$ is an estimator of (label of $\boldsymbol{x}$) $\times$ (label of $\boldsymbol{x}'$) — a quantity indicating whether both features belong to the same class or not — using the dataset $S$.

A natural way to obtain $\mathcal{Z}$ is to form a new *pairwise dataset* $\left\{ \left((\boldsymbol{x}_i, \boldsymbol{x}_j), y_i y_j\right) : i, j \in [n] \right\}$, where we think of $y_i y_j$ as the label of the augmented feature vector $(\boldsymbol{x}_i, \boldsymbol{x}_j)$. If we use a linear regression model, then we would have $\mathcal{Z}(\boldsymbol{x}, \boldsymbol{x}', S) = y^\top \mathbf{M}(\boldsymbol{x}, \boldsymbol{x}', \mathbf{X})y$ for some matrix $\mathbf{M} \in \mathbb{R}^{n \times n}$ depending on the two feature vectors $\boldsymbol{x}, \boldsymbol{x}'$ as well as the feature matrix $\mathbf{X}$, hence the name "higher-order regression".

What requirement we do need to impose on the estimator $\mathcal{Z}$? Heuristically, the inclusion of $\mathcal{Z}$ is only helpful when additional information in the training data is "extracted" apart from the one extracted by the label-agnostic part. We refer the readers to Sec. 3.1 and Appx. D.3 for more details on the choice of $\mathcal{Z}$.

## 2.2 Label-Aware NTK via Approximately Solving NTH

As we have discussed in Section 1, different from the label-agnostic NTK $\mathbb{E}_{\text{init}} K_0^{(2)}$, the time-dependent kernel $K_t^{(2)}$ is indeed label-aware. This suggests that in real-world neural network training, $K_t^{(2)}$ must drift away from $K_0^{(2)}$ by a non-negligible amount. Indeed, Huang & Yau (2019) prove that the second order kernel $K_t^{(2)}$ evolves according to the following infinite *hierarchy* of ordinary differential equations (that is, the NTH):

$$\dot{K}_t^{(r)}(\boldsymbol{x}_{\alpha_1}, \cdots, \boldsymbol{x}_{\alpha_r}) = -\frac{1}{n} \sum_{i \in [n]} K_t^{(r+1)}(\boldsymbol{x}_{\alpha_1}, \cdots, \boldsymbol{x}_{\alpha_r}, \boldsymbol{x}_i)(f_t(\boldsymbol{x}_i) - y_i), \quad r \geq 2, \quad (3)$$

which, along with (1), gives a *precise description* of the training dynamics of NNs. Here, $K_t^{(r)}$ is an *r-th order kernel* which takes $r$ feature vectors as the input.

Our second construction of LANTK is to obtain a finer approximation of $K_t^{(2)}$. Let $\mathbf{f}_t$ be the vector of $f_t(\boldsymbol{x}_i)$'s. Using (3), we can re-write $K_t^{(2)}(\boldsymbol{x}, \boldsymbol{x}')$ as $K_0^{(2)}(\boldsymbol{x}, \boldsymbol{x}') - \frac{1}{n} \int_0^t \langle K_0^{(3)}(\boldsymbol{x}, \boldsymbol{x}', \cdot), \mathbf{f}_u - y \rangle du + \frac{1}{n^2} \int_0^t \int_0^u (\mathbf{f}_u - y)^\top K_v^{(4)}(\boldsymbol{x}, \boldsymbol{x}', \cdot, \cdot)(\mathbf{f}_v - y) dv du$, where $K_t^{(3)}(\boldsymbol{x}, \boldsymbol{x}', \cdot)$ is an $n$-dimensional vector whose $i$-th coordinate is $K_t^{(3)}(\boldsymbol{x}, \boldsymbol{x}', \boldsymbol{x}_i)$, and $K_t^{(4)}(\boldsymbol{x}, \boldsymbol{x}', \cdot, \cdot)$ is an $n \times n$ matrix, whose $(i, j)$-th entry is $K_t^{(4)}(\boldsymbol{x}, \boldsymbol{x}', \boldsymbol{x}_i, \boldsymbol{x}_j)$. Now, let $\mathbf{K}_0^{(2)} \in \mathbb{R}^{n \times n}$ be the kernel matrix corresponding to $K_0^{(2)}$ computed on the training data. The kernel gradient descent using $\mathbf{K}_0^{(2)}$ is characterized by $\dot{\mathbf{h}}_t = -\frac{1}{n} \mathbf{K}_0^{(2)}(\mathbf{h}_t - y)$ with the initial value condition $\mathbf{h}_0 = \mathbf{f}_0$, whose solution is given by $\mathbf{h}_t = (\mathbf{I}_n - e^{-t\mathbf{K}_0^{(2)}/n})y + \mathbf{h}_0$. For a wide enough network, we expect $\mathbf{h}_t \approx \mathbf{f}_t$, at least when $t$ is not too large. On the other hand, it has been shown in Huang & Yau (2019) that $K_t^{(4)}$ varies at a rate of $\tilde{O}(1/m^2)$, where $\tilde{O}$ hides the poly-log factors and $m$ is the (minimum) width of the network. Hence, it is reasonable to approximate $K_t^{(4)}$ by $K_0^{(4)}$. This motivates us to write

$$K_t^{(2)}(\boldsymbol{x}, \boldsymbol{x}') = \hat{K}_t^{(2)}(\boldsymbol{x}, \boldsymbol{x}') + \mathcal{E} \quad (4)$$

where $\mathcal{E}$ is a small error term and $\hat{K}_t^{(2)}(\boldsymbol{x}, \boldsymbol{x}') = K_0^{(2)}(\boldsymbol{x}, \boldsymbol{x}') - \frac{1}{n} \int_0^t \langle K_0^{(3)}(\boldsymbol{x}, \boldsymbol{x}', \cdot), \mathbf{h}_u - y \rangle du + \frac{1}{n^2} \int_0^t \int_0^u (\mathbf{h}_u - y)^\top K_0^{(4)}(\boldsymbol{x}, \boldsymbol{x}', \cdot, \cdot)(\mathbf{h}_v - y) dv du$.

In Appx. A.2, we show that $\hat{K}_t^{(2)}$ can be evaluated analytically, and the analytical expression is of the form $K_0^{(2)} + \mathbf{a}^\top y + y^\top \mathbf{B} y$ for some vector $\mathbf{a}$ and matrix $\mathbf{B}$ (see Proposition A.2 for the exact formula). Moreover, the approximation (4) can be justified formally under mild regularity conditions:

**Theorem 2.1** (Validity of $\hat{K}_t^{(2)}$). *Consider a neural network with $\mathbf{b}^{(\ell)} = 0 \; \forall \ell$ and $p_\ell = q_\ell = m$ except for $q_1 = d, p_L = 1$. We train this net using gradient flow with the least squares loss and Gaussian initialization. Under Assumption A in Appendix A.4, with probability at least $1 - e^{-Cm}$ w.r.t. the initialization, the error of the approximation (4) at time $t$ satisfies*

$$|\mathcal{E}| \lesssim \frac{(\log m)^c}{m^2} \times \left[ t^2(t^2 \vee 1)\left(1 + \frac{t^2}{m}\right)\left(t \wedge \frac{n}{\lambda}\right) + t^3(1 \vee t) \right] \text{ provided } t \lesssim \left(\frac{\sqrt{\lambda m/n}}{(\log m)^{c'}} \wedge \frac{m^{1/3}}{(\log m)^{c''}}\right),$$

*where $C, c, c', c''$ are some absolute constants and $\lambda$ is a quantity defined in Asumption A. On the other hand, on the same high probability event, the three terms in $\hat{K}_t^{(2)}$ are at most $\tilde{O}(1), \tilde{O}(t/m)$ and $\tilde{O}(t^2/m)$, respectively.*

By the above theorem, the error term is much smaller than the main terms for large $m$, justifying the validity of the approximation (4). Meanwhile, this approximation is *strictly better* than $\mathbb{E}_{\text{init}} K_0^{(2)}$, whose error term is shown to be $\tilde{O}(1/m)$ in Huang & Yau (2019).

Based on the approximation (4), we propose the following LANTK-NTH:

$$K^{(\text{NTH})}(\boldsymbol{x}, \boldsymbol{x}') := \mathbb{E}_{\text{init}} K_0^{(2)}(\boldsymbol{x}, \boldsymbol{x}') + y^\top (\mathbb{E}_{\text{init}} \mathbf{K}_0^{(2)})^{-1} \mathbb{E}_{\text{init}}[K_0^{(4)}(\boldsymbol{x}, \boldsymbol{x}', \cdot, \cdot)](\mathbb{E}_{\text{init}} \mathbf{K}_0^{(2)})^{-1} y$$
$$- y^\top \bar{\mathbf{P}} \bar{\mathbf{Q}}(\boldsymbol{x}, \boldsymbol{x}') \bar{\mathbf{P}}^\top y, \quad (5)$$

where $\bar{\mathbf{P}}\bar{\mathbf{D}}\bar{\mathbf{P}}^\top$ is the eigen-decomposition of $\mathbb{E}_{\text{init}}\mathbf{K}_0^{(2)}$ and the $(i,j)$-th entry of $\bar{\mathbf{Q}}(\boldsymbol{x}, \boldsymbol{x}')$ is given by $\big(\bar{\mathbf{P}}^\top \mathbb{E}_{\text{init}}[K_0^{(4)}(\boldsymbol{x}, \boldsymbol{x}', \cdot, \cdot)]\bar{\mathbf{P}}\bar{\mathbf{D}}^{-1}\big)_{ij}/(\bar{\mathbf{D}}_{ii} + \bar{\mathbf{D}}_{jj})$. In a nutshell, we take the formula for $\hat{K}^{(2)}$, integrate it w.r.t. the initialization, and send $t \to \infty$. The linear term $\mathbf{a}^\top y$ vanishes as $\mathbb{E}_{\text{init}}[\mathbf{a}] = 0$.[2]

## 2.3 Understanding the connection between LANTK-HR and LANTK-NTH

The formulas for the two LANTKs are perhaps unexpectedly similar — they are both quadratic functions of $y$ (here we focus on linear regression based $\mathcal{Z}$ in $K^{(\text{HR})}$). Is there any intrinsic relationship between the two? In this section, we propose to understand their relationship via a classical tool from asymptotic statistics called *Hoeffding decomposition* (Hoeffding et al., 1948). Let us start by considering a generic label-aware kernel $K(\boldsymbol{x}, \boldsymbol{x}') \equiv K(\boldsymbol{x}, \boldsymbol{x}', S)$, which may have arbitrary dependence on the training data $S$. For a fixed $A \subseteq [n]$, define

$$\mathcal{G}_A := \left\{ g : y_A \mapsto f(y_A) \;\middle|\; \mathbb{E}_{y|\mathbf{X}} g(y_A)^2 < \infty, \mathbb{E}_{y|\mathbf{X}}[g(y_A)|y_B] = 0 \; \forall B \text{ s.t. } |B| < |A| \right\}, \quad (6)$$

The requirement of $\mathbb{E}_{y|\mathbf{X}}[g(y_A)|y_B] = 0 \; \forall B$ s.t. $|B| < |A|$ means that for any $B$ whose cardinality is less than $A$, the function $g$ has no information in $y_B$, which reflects our intention to orthogonally decompose a generic function of $y$ into parts whose dependence on $y$ is increasingly complex.

The celebrated work of Hoeffding et al. (1948) tells that the function spaces $\{\mathcal{G}_A : A \subseteq [n]\}$ form an orthogonal decomposition of $\sigma(y \mid \mathbf{X})$, the space of all measurable functions of $y$ (w.r.t. the conditional law of $y \mid \mathbf{X}$). Specifically, we have:

**Lemma 2.2** (Hoeffding decomposition of a generic label-aware kernel). *We can decompose a generic label-aware $K$ into*

$$K(\boldsymbol{x}, \boldsymbol{x}', S) = \sum_{A \subseteq [n]} \text{proj}_A K(\boldsymbol{x}, \boldsymbol{x}', S) = \sum_{r=0}^{n} \sum_{A \subseteq [n]:|A|=r} \text{proj}_A K(\boldsymbol{x}, \boldsymbol{x}', S), \quad (7)$$

*where $\text{proj}_A$ is the $L^2$ projection operator onto the function space $\mathcal{G}_A$. In particular, we can write*

$$K(\boldsymbol{x}, \boldsymbol{x}', S) = \mathcal{K}_\varnothing^{(2)}(\boldsymbol{x}, \boldsymbol{x}') + \sum_{i \in [n]} \mathcal{K}_i^{(3)}(\boldsymbol{x}, \boldsymbol{x}', y_i) + \sum_{i \neq j} \mathcal{K}_{ij}^{(4)}(\boldsymbol{x}, \boldsymbol{x}', y_i, y_j) + \cdots, \quad (8)$$

*where $\{\mathcal{K}_A^{(r+2)} : 0 \le r \le n, |A| = r\}$ is a collection of measurable functions s.t. $\mathcal{K}_A^{(r+2)}$ only depends on $y_A$.*

Thus, we have decomposed a generic label-aware kernel into the superimposition of a hierarchy of smaller kernels with *increasing complexity* on the label dependence structure: at the zero-th level, $\mathcal{K}_\varnothing^{(2)}$ is label-agnostic; at the first level, $\{\mathcal{K}_i^{(3)}\}$ depend only on a single coordinate of $y$; at the second level, $\{\mathcal{K}_{ij}^{(4)}\}$ depend on a pair of labels, etc. Moreover, *the information at each level is orthogonal.*

In view of the above lemma, the two LANTKs we proposed can be regarded as *truncated versions* of (8), where the truncation happens at the second level.

The above observation motivates us to ask a more "fine-grained" question regarding the two LANTKs: *is the label-aware part in $K^{(\text{NTH})}$ implicitly doing a higher-order regression?*

To offer some intuitions for this question, we randomly choose 5 planes and 5 ships in CIFAR-10[3], and we compute the intra-class and inter-class values of the label-aware component in a two-layer fully-connected $K^{(\text{NTH})}$ (See Appx. B for the exact formulas). We find that the mean of the intra-class values is 98, whereas the mean of the inter-class values is 48. The difference between the two means indicates that $K^{(\text{NTH})}$ may implicitly try to increase the similarity between two examples if they come from the same class and decrease it otherwise, and this behavior agrees with the *intention* of $K^{(\text{HR})}$.

We conclude this section by remarking that our above arguments are, of course, largely heuristic, and a rigorous study on the relationship between the two LANTKs is left as future work.

|  | deer vs dog | cat vs deer | cat vs frog | deer vs frog | bird vs frog | Avg | Imp. (abs./rel.) |
|---|---|---|---|---|---|---|---|
| CNTK | 85.15 | 83.55 | 86.95 | 87.55 | 86.35 | 85.91 | - |
| LANTK-best | **86.75** | **84.85** | **87.40** | **88.30** | **87.45** | **86.95** | 1.04 / 7.4% |
| LANTK-KR-V1 | 85.65 | 83.90 | 87.20 | 87.85 | **87.45** | 86.41 | 0.50 / 3.5% |
| LANTK-KR-V2 | 85.80 | 83.90 | 87.15 | 87.90 | 87.25 | 86.40 | 0.49 / 3.5% |
| LANTK-FJLT-V1 | **86.75** | **84.85** | **87.40** | 88.10 | 86.40 | **86.70** | 0.79 / 5.6% |
| LANTK-FJLT-V2 | 86.25 | 84.55 | 87.10 | **88.30** | 87.00 | 86.64 | 0.73 / 5.2% |
| CNN | 89.00 | 88.50 | 89.00 | 92.50 | 90.15 | 89.83 | 3.92 / 27.8% |

Table 1: Performance of LANTK , CNTK and CNN on binary image classification tasks on CIFAR-10. Note that the best LANTK(indicated as LANTK-best) significantly outperforms CNTK. Here, "abs" stands for absolute improvement and "rel" stands for the relative error reduction.

| Training size | 2000 | 5000 | 10000 | 20000 | Avg | Imp. (abs./rel.) |
|---|---|---|---|---|---|---|
| CNTK | 44.01 | 50.44 | 55.18 | 60.12 | 52.44 | - |
| LANTK-best | **46.31** | **52.66** | **56.96** | **61.58** | **54.38** | 1.94 / 4.1% |
| LANTK-KR-V1 | 44.49 | 51.39 | 55.74 | 60.87 | 53.12 | 0.68 / 1.4% |
| LANTK-KR-V2 | 44.64 | 51.38 | 55.89 | 60.87 | 53.20 | 0.76 / 1.6% |
| LANTK-FJLT-V1 | **46.31** | **52.66** | **56.96** | **61.58** | **54.38** | 1.94 / 4.1% |
| LANTK-FJLT-V2 | 45.72 | 52.50 | 56.26 | Failed | - | 1.62 / 3.4% |
| CNN | 45.30 | 52.70 | 60.23 | 68.15 | 56.60 | 4.16 / 8.7% |

Table 2: Performance of LANTK , CNTK, and CNN on multi-class image classification tasks on CIFAR-10. The improvement is more evident than the binary classification.

# 3   Experiments

Though we have proved in Theorem 2.1 that $K^{(\text{NTH})}$ possesses favorable theoretical guarantees, it's computational cost is prohibitive for large-scale experiments. Hence, in the rest of this section, all experiments on LANTKs are based on $K^{(\text{HR})}$. However, in view of their close connections established in Sec. 2.3, we conjecture that similar results would hold for $K^{(\text{NTH})}$.

## 3.1   Generalization Ability

We compare the generalization ability of the LANTK to its label-agnostic counterpart in both binary and multi-class image classification tasks on CIFAR-10. We consider an 8-layer CNN and its corresponding CNTK. The implementation of CNTK is based on Novak et al. (2020) and the details of the architecture can be found in Appx. D.2.

**Choice of higher-order regression methods.** Due to the high computational cost of kernel methods, our choice of $\mathcal{Z}$ is particularly simple. We consider two methods: one is a kernelized regression (LANTK-KR-V1 and V2[4]), and the other is a linear regression with Fast-Johnson-Lindenstrauss-Transform (LANTK-FJLT-V1 and V2), a sketching method that accelerates the computation (Ailon & Chazelle, 2009). We refer the readers to Appx. D.3 for further details. The choice of "fancier" $\mathcal{Z}$ is deferred to future work.

**Binary classification.** We first choose five pairs of categories on which the performance of the 2-layer fully-connected NTK is neither too high (o.w. the improvement will be marginal) nor too low (o.w. it may be difficult for $\mathcal{Z}$ to extract useful information). We then randomly sample 10000 examples as the training data and another 2000 as the test data, under the constraint that the sizes of positive and negative examples are equal. The performance on the five binary classification tasks are shown in Table 1. We see that LANTK-FJLT-V1 works the best overall, followed by LANTK-FJLT-V2 (which uses more features). The improvement compared to CNTK indicates the importance of label information and confirms our claim that LANTK better simulates the quantitative behaviors of NNs.

**Multi-class classification.** Our current construction of LANTK is under binary classification tasks, but it can be easily extended to multi-class classification tasks by changing the definition of the "optimal" kernel to $K^{\star}(\boldsymbol{x}_{\alpha}, \boldsymbol{x}_{\beta}) = \mathbf{1}\{y_{\alpha} = y_{\beta}\}$. Here, we again samples from CIFAR-10 under the constraint that the classes are balanced, and we vary the size of the training data in $\{2000, 5000, 10000, 20000\}$ while fixing the size of the test data to be 10000. The results are shown

| Train-train | frog vs ship | frog vs truck | deer vs ship | dog vs truck | bird vs truck | deer vs truck |
|---|---|---|---|---|---|---|
| NN-init | 58.37 | 55.07 | 57.50 | 54.75 | 52.93 | 54.86 |
| NN-trained | **71.99** ↑ | **68.36** ↑ | **69.98** ↑ | **66.35** ↑ | **63.99** ↑ | **65.96** ↑ |
| NTK | 63.83 | 58.31 | 62.43 | 58.05 | 55.01 | 58.02 |
| LANTK | **66.62** ↑ | **60.57** ↑ | **64.90** ↑ | **59.75** ↑ | **55.94** ↑ | **59.59** ↑ |

| Test-train | frog vs ship | frog vs truck | deer vs ship | dog vs truck | bird vs truck | deer vs truck |
|---|---|---|---|---|---|---|
| NN-init | 58.31 | 55.06 | 57.64 | 54.62 | 52.93 | 54.94 |
| NN-trained | **71.45** ↑ | **67.91** ↑ | **69.73** ↑ | **65.80** ↑ | **63.58** ↑ | **65.53** ↑ |
| NTK | 63.76 | 58.30 | 62.67 | 57.84 | 55.00 | 58.14 |
| LANTK | **66.53** ↑ | **60.08** ↑ | **65.20** ↑ | **59.54** ↑ | **55.97** ↑ | **59.77** ↑ |

Table 3: Strength of local elasticity in binary classification tasks on CIFAR-10. The training makes NNs more locally elastic, and LANTK successfully simulates this behavior.

in Fig. 2. "Failed" in the table means that the experiment is failed due to memory constraint. Similarly, LANTK-FJLT-V1 works the best among all training sizes, and the performance gain is even higher compared to binary classification tasks. This supports our claim that the label information is particularly important when the semantic meanings of the features are highly dependent on the label system.

## 3.2 Local Elasticity

Local elasticity, originally proposed by He & Su (2020), refers to the phenomenon that the prediction of a feature vector $x'$ by a classifier (not necessarily a NN) is not significantly perturbed, after this classifier is updated via stochastic gradient descent at another feature vector $x$, if $x$ and $x'$ are *dissimilar* according to some geodesic distance which captures the semantic meaning of the two feature vectors. Through extensive experiments, He & Su (2020) demonstrate that NNs are locally elastic, whereas linear classifiers are not. Moreover, they show that models trained by NTK is far less locally elastic compared to NNs, but more so compared to linear models.

In this section, we show that LANTK is significantly more locally elastic than NTK. We follow the experimental setup in He & Su (2020). Specifically, for a kernel $K$, define the *normalized kernelized similarity*[5] as $\bar{K}(x, x') := K(x, x')/\sqrt{K(x, x)K(x', x')}$. Then, the strength of local elasticity of the corresponding kernel regression can be quantified by the relative ratio of $\bar{K}$ between intra-class examples and inter-class examples:

$$\text{RR}(K) := \frac{\sum_{(x,x')\in S_1} \bar{K}(x, x')/|S_1|}{\sum_{(x,x')\in S_1} \bar{K}(x, x')/|S_1| + \sum_{(\tilde{x},\tilde{x}')\in S_2} \bar{K}(\tilde{x}, \tilde{x}')/|S_2|}, \tag{9}$$

where $S_1$ is the set of intra-class pairs and $S_2$ is the set of inter-class pairs. Note that under this setup, the strength of local elasticity of NNs corresponds to $\text{RR}(K_t^{(2)})$. In practice, we can take the pairs to both come from the training set (train-train pairs), or one from the test set and another from the training set (test-train pairs).

We first compute the strength of local elasticity for two-layer NNs with $40960$ hidden neurons at initialization and after training. We find that for all $\binom{10}{2} = 45$ binary classification tasks in CIFAR-10, RR significantly increases after training, under both train-train and test-train settings, agreeing with the result in Fig. 1(b) and Fig. 1(c). The top 6 tasks with the largest increase in RR in Table 3. The absolute improvement in the test accuracy is shown in Fig. 1(a).

We then compute the strength of local elasticity for the corresponding NTK and LANTK (specifically, LANTK-KR-V1) based on the formulas derived in Appx. B.3.1, and the results for the same 6 binary classification tasks are shown in Table 3. We find that LANTK is more locally elastic than NTK, indicating that LANTK better simulates the qualitative behaviors of NNs.

## 4 Conclusion

In this paper, we proposed the notion of label-awareness to explain the performance gap between a model trained by NTK and real-world NNs. Inspired by the Hoeffding Decomposition of a generic

label-aware kernel, we proposed two label-aware versions of NTK, both of which are shown to better simulate the behaviors of NNs via a theoretical study and comprehensive experiments.

We conclude this paper by mentioning several potential future directions.

**More efficient implementations.** Our implementation of $K^{\text{(HR)}}$ requires forming a pairwise dataset, which can be cumbersome in practice (indeed, we need to use fast FJLT to accelerate the computation). Moreover, the exact computation of $K^{\text{(NTH)}}$ requires at least $O(n^4)$ time, since the dimension of the matrix $\mathbb{E}_{\text{init}}[K_0^{(4)}(\boldsymbol{x}, \boldsymbol{x}', \cdot, \cdot)]$ is $n^2 \times n^2$. It would greatly improve the practical usage of our proposed kernels if there are more efficient implementations.

**Higher-level truncations.** As discussed in Sec. 2.3, our proposed $K^{\text{(HR)}}$ and $K^{\text{(NTH)}}$ can be regarded as *second-level truncations* of the Hoeffding Decomposition (7). In principle, our constructions can be generalized to higher-level truncations, which may give rise to even better "NN-simulating" kernels. However, such generalizations would incur even higher computational costs. It would be interesting to see even such generalizations can be done with a reasonable amount of computational resources.

**Going beyond least squares and gradient flows.** Our current derivation is based on a neural network trained by squared loss and gradient flows. While such a formulation is common in the NTK literature and makes the theoretical analysis simpler, it is of great interest to extend the current analysis to more practical loss functions and optimization algorithms.

## Acknowledgments

This work was in part supported by NSF through CAREER DMS-1847415 and CCF-1934876, an Alfred Sloan Research Fellowship, the Wharton Dean's Research Fund, and Contract FA8750-19-2-0201 with the US Defense Advanced Research Projects Agency (DARPA).

## Broader Impact

While this work may have certain implications on the design and analysis of new kernel methods, here we focus on how this work can potentially influence the interpretation of deep learning systems. In real-world decision-making problems, interpretability is almost always a crucial factor to consider if one is to deploy a machine learning system. For example, in autonomous driving where NNs are used to detect pedestrians and traffic lights, it is important to understand why this detection network outputs a certain prediction and how confident it is for such a prediction, lack of which can cause damages to the surrounding pedestrians and other drivers. Such a call for interpretability is underlying many works on the "calibration" of NNs (see, e.g., Guo et al. 2017).

Kernel methods, due to its linearity in the feature space, are easier to interpret than highly non-linear NNs, which is typically treated as a black-box. Thus, having a high-quality "neural-network-simulating" kernel can greatly simplify the design of "neural network interpreters" (like prediction intervals) and may lead to savings of computational resources. However, depending on the user of our technology, there may be negative outcomes. For example, if the user is ignorant of the underlying assumptions behind the validity of our proposed kernels, he/she may have an overt optimism or undue trust on these kernels and make misleading decisions.

We see many potential research directions on improving neural network interpretability by using our kernels. For example, our constructions can be generalized to higher-level truncations of the Hoeffding composition, which may give rise to even better "neural-network-simulating" kernels. However, to mitigate the risks associated with the question of "when a kernel is indeed simulating a neural network", we encourage researchers to carefully examine the validity of the imposed assumptions in a case-by-case manner.

## Footnotes

[1]Our code is publicly available at `https://github.com/HornHehhf/LANTK`.

[2]We can in principle prove a similar result as Theorem 2.1 for $K^{(\text{NTH})}$ by exploiting the concentration of $K_0^{(3)}$ and $K_0^{(4)}$, but since it is not the focus of this paper, we omit the details.

[3]We only sample such a small number of images because the computation cost of $K^{(\text{NTH})}$ is at least $O(n^4)$.

[4]V1 and V2 stand for two ways of extracting features.

[5]This similarity measure is closely related stiffness (Fort et al., 2019), but the difference lies in that stiffness takes the loss function into account.

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
