[Supplementary Material]



Figure 2: Pictorial illustration of the example that validates Claim 1.1. The black "+" and "−" represent positive and negative examples according to the label system induced by $\boldsymbol{w}_1$, and the solid line represents the corresponding separating hyperplane. The dashed line represents the hyperplane whose normal vector is $\boldsymbol{w}_2$, which, along with the solid line, partition that $\phi$ space into four quadrants. The red "+" and "−" represents the positive and negative examples under the new label system.

## A  Technical Details

### A.1  A Example Validating Claim 1.1

We first formally define the notion of a "label system":

**Definition A.1** (Label system). *Let $(X, Y)$ be a fresh sample from the data distribution. The label system is the function $\eta : \mathbb{R}^d \to [0, 1]$ defined by*

$$\eta(\boldsymbol{x}) = \mathbb{P}(Y = 1 \mid X = \boldsymbol{x}).$$

Let $\phi(\cdot)$ be the feature map corresponding to $K$, so that $K(\boldsymbol{x}, \boldsymbol{x}') = \langle \phi(\boldsymbol{x}), \phi(\boldsymbol{x}') \rangle$, where $\langle \cdot, \cdot \rangle$ is the inner product in the $\phi$-space. By our assumption, $\phi$ is also label-agnostic.

Suppose $\phi$ works well on the label system $\eta_1$. This means that a simple linear fit in the $\phi$ space suffices to achieve satisfactory accuracy. Geometrically, this translates to the existence of a hyperplane which can almost perfectly separate the two classes. In other words, we can find a normal vector $\boldsymbol{w}_1$[6], such that $\mathbb{P}_{\eta_1}\left(Y \cdot \langle \boldsymbol{w}_1, \phi(X) \rangle > 0\right) \approx 1$, where we use $\mathbb{P}_{\eta_1}$ to stress that $Y$ comes from the label system $\eta_1$.

Now, choose any vector $\boldsymbol{w}_2$ in the $\phi$ space, such that $\boldsymbol{w}_2$ is orthogonal to $\boldsymbol{w}_1$. We consider the following relabelling procedure, which produces another label system $\eta_2$:

$$\eta_2(\boldsymbol{x}) = \begin{cases} 1 & \text{if } \langle \boldsymbol{w}_1, \phi(\boldsymbol{x}) \rangle \cdot \langle \boldsymbol{w}_2, \phi(\boldsymbol{x}) \rangle < 0 \\ 0 & \text{otherwise.} \end{cases} \tag{10}$$

Under $\eta_2$, the $\phi$-space is partitioned into four quadrants, which we label as I, II, III, IV counterclockwise. Then, any $\boldsymbol{x}$ in I and III is labelled as $-1$, and any $\boldsymbol{x}$ in II and IV is labelled as $+1$. Obviously, there is no hyper-plane that can separate the $+1$ and $-1$ examples, meaning that $\phi$ can be arbitrarily bad under $\eta_2$, hence validating Claim. See Fig. 2 for a pictorial illustration.

### A.2  Exact integrability of the approximate $K_t^{(2)}$

In this subsection, we prove the following result:

**Proposition A.2** (Exact integrability of the approximate $K_t^{(2)}$). *Assume* $\mathbf{f}_0 = \mathbf{0}_n$ [7], *then the first three terms in RHS of* (4) *is equal to*

$$K_0^{(2)}(\boldsymbol{x}, \boldsymbol{x}') + \left\langle K_0^{(3)}(\boldsymbol{x}, \boldsymbol{x}', \cdot), (\mathbf{K}_0^{(2)})^{-1}\left(\mathbf{I}_n - e^{-t\mathbf{K}_0^{(2)}/n}\right)y \right\rangle$$

$$+ y^\top (\mathbf{K}_0^{(2)})^{-1}\left(\mathbf{I}_n - e^{-t\mathbf{K}_0^{(2)}/n}\right)K_0^{(4)}(\boldsymbol{x}, \boldsymbol{x}', \cdot, \cdot)(\mathbf{K}_0^{(2)})^{-1}y - y^\top \mathbf{P}\mathbf{Q}(\boldsymbol{x}, \boldsymbol{x}')\mathbf{P}^\top y,$$

*where*

$$\left(\mathbf{Q}(\boldsymbol{x}, \boldsymbol{x}')\right)_{ij} = (1 - e^{-t(\mathbf{D}_{ii}+\mathbf{D}_{jj})/n}) \times \left(\mathbf{P}^\top K_0^{(4)}(\boldsymbol{x}, \boldsymbol{x}', \cdot, \cdot)\mathbf{P}\mathbf{D}^{-1}\right)_{ij} \Big/ \left(\mathbf{D}_{ii} + \mathbf{D}_{jj}\right),$$

*and* $\mathbf{P}\mathbf{D}\mathbf{P}^\top$ *is the eigen-decomposition of* $\mathbf{K}_0^{(2)}$.

*Proof.* For notational simplicity, we let $\mathbf{H} \equiv \mathbf{K}_0^{(2)}$. We have

$$\int_0^t \mathbf{h}_u - y du = \int_0^t e^{-u\mathbf{H}/n}y du = -n\mathbf{H}^{-1}y + n\mathbf{H}^{-1}e^{-t\mathbf{H}/n}y.$$

Hence, the second term in the RHS of (4) is

$$-\frac{1}{n}\int_0^t \left\langle K_0^{(3)}, \mathbf{h}_u - y \right\rangle du = \left\langle K_0^{(3)}, \mathbf{H}^{-1}\left(\mathbf{I}_n - e^{-t\mathbf{H}/n}\right)y \right\rangle.$$

We now deal with the third term in the RHS of (4). We have

$$\int_0^t \int_0^u (\mathbf{h}_u - y)^\top K_0^{(4)}(\mathbf{h}_v - y)dvdu$$

$$= y^\top \int_0^t \int_0^u e^{-u\mathbf{H}/n}K_0^{(4)}e^{-v\mathbf{H}/n}ydvdu$$

$$= y^\top \int_0^t e^{-u\mathbf{H}/n}K_0^{(4)}(n\mathbf{H}^{-1} - n\mathbf{H}^{-1}e^{-u\mathbf{H}/n})y$$

$$= n^2 y^\top \mathbf{H}^{-1}\left(\mathbf{I}_n - e^{-t\mathbf{H}/n}\right)K_0^{(4)}\mathbf{H}^{-1}y - ny^\top \int_0^t e^{-u\mathbf{H}/n}K_0^{(4)}\mathbf{H}^{-1}e^{-u\mathbf{H}/n}ydu.$$

Let $\mathbf{P}\mathbf{D}\mathbf{P}^\top$ be an eigen-decomposition of $\mathbf{H}$. Then we have

$$\int_0^t e^{-u\mathbf{H}/n}K_0^{(4)}\mathbf{H}^{-1}e^{-u\mathbf{H}/n}du = \int_0^t \mathbf{P}e^{-u\mathbf{D}/n}\underbrace{\mathbf{P}^\top K_0^{(4)}\mathbf{P}\mathbf{D}^{-1}}_{:=\mathbf{M}}e^{-u\mathbf{D}/n}\mathbf{P}^\top du$$

$$= \mathbf{P}\int_0^t e^{-t\mathbf{D}/n}\mathbf{M}e^{-\mathbf{D}/n}du\mathbf{P}^\top.$$

Note that

$$\left(e^{-t\mathbf{D}/n}\mathbf{M}e^{-\mathbf{D}/n}\right)_{ij} = \mathbf{M}_{ij}e^{-u(d_i+d_j)/n},$$

here $d_i := \mathbf{D}_{ii}$. Hence,

$$\int_0^t \left(e^{-t\mathbf{D}/n}\mathbf{M}e^{-\mathbf{D}/n}\right)_{ij} du = \mathbf{M}_{ij} \cdot \left(\frac{n}{d_i + d_j} - \frac{n}{d_i + d_j}e^{-t(d_i+d_j)/n}\right),$$

which yields

$$\int_0^t e^{-u\mathbf{H}/n}K_0^{(4)}\mathbf{H}^{-1}e^{-u\mathbf{H}/n}du = n\mathbf{P}\mathbf{Q}\mathbf{P}^\top.$$

Putting the above equations together gives the desired result. □

## A.3 Proof of Lemma 2.2

Note that the finite variance assumption in the definition of $\mathcal{G}_A$ is vacuous, because our label is binary. The rest of the proof is standard (see, e.g., Section 11.4 of Van der Vaart 2000).

## A.4 Proof of Theorem 2.1

**Assumption A.** *There exists a small constant $c > 0$ such that $c < \|\boldsymbol{x}_i\|_2 \leq c^{-1}$ for all $i \in [n]$. Moreover, there exists an integer $p \geq 4$ such that for any $1 \leq r \leq 2p + 1$, the following two things happen:*

1. *the activation function has a bounded $r$-th derivative;*

2. *there exists a constant $c_r > 0$ such that for any distinct indices $1 \leq \alpha_1, \alpha_2, \cdots, \alpha_r \leq n$, the smallest singular value of the data matrix $[\boldsymbol{x}_{\alpha_1}, \boldsymbol{x}_{\alpha_2}, \cdots, \boldsymbol{x}_{\alpha_r}]$ is at least $c_r$.*

To prove Theorem 2.1, we first collect some useful details into the following lemma.

**Lemma A.3.** *Under the assumptions of Theorem 2.1, with high probability, for any $u \leq t$, we have*

$$\|K_u^{(3)}\|_\infty = \tilde{O}\left(\frac{1+u}{m}\right), \quad \|K_u^{(4)}\|_\infty = \tilde{O}\left(\frac{1}{m}\right), \quad \|K_u^{(5)}\|_\infty = \tilde{O}\left(\frac{1+u}{m^2}\right),$$

$$\|\mathbf{f}_u - \mathbf{h}_u\|_2 \lesssim \frac{u(1+u)\sqrt{n}}{m}\left(u \wedge \frac{n}{\lambda}\right), \quad \|\mathbf{f}_u - y\|_2 = O(\sqrt{n}).$$

*Proof.* This is implied by Equations (C.12), (C.13), (C.16), and (C.28) in Huang & Yau (2019). $\quad\square$

We are now ready to present the proof.

*Proof of Theorem 2.1.* We have

$$K_t^{(2)}(\boldsymbol{x}_\alpha, \boldsymbol{x}_\beta)$$

$$= K_0^{(2)}(\boldsymbol{x}_\alpha, \boldsymbol{x}_\beta) - \frac{1}{n}\left\langle K_0^{(3)}, \int_0^t \mathbf{h}_u - y + \mathbf{f}_u - \mathbf{h}_u du \right\rangle$$

$$+ \frac{1}{n^2}\int_0^t \int_0^u (\mathbf{h}_u - y + \mathbf{f}_u - \mathbf{h}_u)^\top (K_0^{(4)} + K_v^{(4)} - K_0^{(4)})(\mathbf{h}_v - y + \mathbf{f}_v - \mathbf{h}_v)dvdu$$

$$= K_0^{(2)}(\boldsymbol{x}_\alpha, \boldsymbol{x}_\beta) - \frac{1}{n}\left\langle K_0^{(3)}, \int_0^t \mathbf{h}_u - ydu \right\rangle + \frac{1}{n^2}\int_0^t \int_0^u (\mathbf{h}_u - y)^\top K_0^{(4)}(\mathbf{h}_v - y)dvdu + \mathcal{E},$$

where the error term $\mathcal{E}$ is given by

$$\mathcal{E} = -\frac{1}{n}\left\langle K_0^{(3)}, \int_0^t \mathbf{f}_u - \mathbf{h}_u du \right\rangle + \frac{1}{n^2}\int_0^t \int_0^u (\mathbf{h}_u - y)^\top K_0^{(4)}(\mathbf{f}_v - \mathbf{h}_v)dvdu$$

$$+ \frac{1}{n^2}\int_0^t \int_0^u (\mathbf{h}_u - y)^\top (K_v^{(4)} - K_0^{(4)})(\mathbf{f}_v - y)dvdu$$

$$+ \frac{1}{n^2}\int_0^t \int_0^u (\mathbf{f}_u - \mathbf{h}_u)^\top K_v^{(4)}(\mathbf{f}_v - y)dvdu.$$

We label the four terms in the RHS above as I, II, III, and IV. By lemma A.3, for the first term, w.h.p. we have

$$\text{I} \leq \frac{1}{n}\|K_0^{(3)}(\boldsymbol{x}_\alpha, \boldsymbol{x}_\beta, \cdot)\|_2 \cdot \int_0^t \|\mathbf{f}_u - \mathbf{h}_u\|_2 du$$

$$\lesssim \frac{1}{m}\|K_0^{(3)}\|_\infty \cdot \int_0^t u(1+u)\left(u \wedge \frac{n}{\lambda}\right)du$$

$$\lesssim \frac{(\log m)^c}{m^2} \cdot \left((t^3 + t^4) \wedge \frac{n(t^2 + t^3)}{\lambda}\right)$$

$$\lesssim \frac{(\log m)^c \cdot t^2(1+t)}{m^2}\left(t \wedge \frac{n}{\lambda}\right).$$

For the second term, we have

$$\text{II} \leq \frac{1}{n^2} \int_0^t \int_0^u \|\mathbf{h}_u - y\|_2 \|\mathbf{f}_v - \mathbf{h}_v\|_2 \|K_0^{(4)}(\boldsymbol{x}_\alpha, \boldsymbol{x}_\beta, \cdot, \cdot)\| dv du$$

$$\lesssim \frac{1}{m} \int_0^t \int_0^u v(1+v)\left(v \wedge \frac{n}{\lambda}\right) \|K_0^{(4)}\|_\infty dv du$$

$$\lesssim \frac{(\log m)^c}{m^2} \int_0^t \int_0^u v(1+v)\left(v \wedge \frac{n}{\lambda}\right)$$

$$\lesssim \frac{(\log m)^c}{m^2}(t^4 + t^5) \wedge \frac{n(t^3 + t^4)}{\lambda}$$

$$= \frac{(\log m)^c \cdot t^3(1+t)}{m^2}\left(t \wedge \frac{n}{\lambda}\right).$$

For the third term, we have

$$\text{III} \leq \frac{1}{n^2} \int_0^t \int_0^u \|\mathbf{h}_u - y\|_2 \|\mathbf{f}_v - y\|_2 \|K_v^{(4)}(\boldsymbol{x}_\alpha, \boldsymbol{x}_\beta, \cdot, \cdot) - K_0^{(4)}(\boldsymbol{x}_\alpha, \boldsymbol{x}_\beta, \cdot, \cdot)\|_2 dv du.$$

Note that

$$\|\mathbf{h}_u - y\|_2 \leq \|\mathbf{h}_u - \mathbf{f}_u\|_2 + \|\mathbf{f}_u - y\|_2 \lesssim \frac{u(1+u)\sqrt{n}}{m}\left(u \wedge \frac{n}{\lambda}\right) + \sqrt{n},$$

$$\|\mathbf{f}_v - y\|_2 \lesssim \sqrt{n},$$

and

$$\|K_v^{(4)}(\boldsymbol{x}_\alpha, \boldsymbol{x}_\beta, \cdot, \cdot) - K_0^{(4)}(\boldsymbol{x}_\alpha, \boldsymbol{x}_\beta, \cdot, \cdot)\|_2$$

$$\leq n \max_{i,j \in [n]} |K_v^{(4)}(\boldsymbol{x}_\alpha, \boldsymbol{x}_\beta, \boldsymbol{x}_i, \boldsymbol{x}_j) - K_0^{(4)}(\boldsymbol{x}_\alpha, \boldsymbol{x}_\beta, \boldsymbol{x}_i, \boldsymbol{x}_j)|$$

$$\leq n \int_0^v \max_{i,j \in [n]} |\dot{K}_w^{(4)}(\boldsymbol{x}_\alpha, \boldsymbol{x}_\beta, \boldsymbol{x}_i, \boldsymbol{x}_j)| dw$$

$$= n \int_0^v \max_{i,j \in [n]} \frac{1}{n}\left| \sum_{k \in [n]} K_w^{(5)}(\boldsymbol{x}_\alpha, \boldsymbol{x}_\beta, \boldsymbol{x}_i, \boldsymbol{x}_j, \boldsymbol{x}_k)(f_w(\boldsymbol{x}_k) - y_k)\right| dw$$

$$\leq n \int_0^v \|K_w^{(5)}\|_\infty \frac{1}{\sqrt{n}} \|\mathbf{f}_w - y\|_2 dw$$

$$\lesssim \frac{n(\log m)^c}{m^2} \int_0^v (1+w) dw$$

$$\lesssim \frac{n(\log m)^c \cdot (v + v^2)}{m^2}.$$

Hence, we can bound the third term by

$$\text{III} \lesssim \frac{(\log m)^c}{m^2} \int_0^t \int_0^u (v + v^2)\left(1 + \frac{u(1+u)(u \wedge n/\lambda)}{m}\right) dv du$$

$$\lesssim \frac{(\log m)^c \cdot t^3(1+t)}{m^2} + \frac{(\log m)^c \cdot t^4(1+t+t^2)}{m^3}\left(t \wedge \frac{n}{\lambda}\right).$$

Finally, we bound the fourth term by

$$\text{IV} \leq \frac{1}{n^2} \int_0^t \int_0^u \|\mathbf{f}_u - \mathbf{h}_u\|_2 \|\mathbf{f}_v - y\|_2 \|K_v^{(4)}(\boldsymbol{x}_\alpha, \boldsymbol{x}_\beta, \cdot, \cdot)\|_2 dv du$$

$$\lesssim \frac{(\log m)^c}{m^2} \int_0^t \int_0^u u(1+u)\left(u \wedge \frac{n}{\lambda}\right) dv du$$

$$\lesssim \frac{(\log m)^c \cdot t^3(1+t)}{m^2}\left(t \wedge \frac{n}{\lambda}\right).$$

Combining the above four bounds gives the desired bound on $\mathcal{E}$. The bounds for the three terms in the RHS of (4) are derived similarly, and we omit the details. $\square$

## B NTH-Related Calculations

### B.1 An Symbolic Program to Compute NTH

In this section, we develop a recursive program to symbolically compute $K_t^{(r)}$, the $r$-th order kernel in NTH. For simplicity, we consider neural nets without biases [8]:

$$f(\boldsymbol{x}, \boldsymbol{w}) = \mathbf{W}^{(L)} \sigma\bigg( \mathbf{W}^{(L-1)} \cdots \big( \mathbf{W}^{(2)} \sigma(\mathbf{W}^{(1)}\boldsymbol{x}) \big) \bigg).$$

We begin by noting that $K_t^{(2)}$ can be written as an inner product of two gradients. We will see that $K_t^{(3)}$ can be written as a quadratic form, and $K^{(4)}$ can be written as a cubic form, etc.

To this end, let us denote $\nabla f(\boldsymbol{x}, \boldsymbol{w})$ to be the partial derivative of $f(\boldsymbol{x}, \boldsymbol{w})$ w.r.t. $\boldsymbol{w}$. We sometimes drop the dependence on $t$ and write $f(\boldsymbol{x}_\alpha, \boldsymbol{w}_t) \equiv f_\alpha$ when there is no ambiguity.

We will regard $\boldsymbol{w} = \mathbf{W} = \{\mathbf{W}_{jk}^{(\ell)} : \ell \in [L], j \in [p_\ell], j \in [q_\ell]\}$ as a rank-3 tensor. We write $\mathbf{W}_{jk}^{(\ell)} \equiv \mathbf{W}_{\ell jk}$ when there is no ambiguity. In our current notations, $\nabla f(\boldsymbol{x}, \boldsymbol{w}) = \{\big(\nabla f(\boldsymbol{x}, \boldsymbol{w})\big)_{\ell jk} : \ell \in [L], j \in p_\ell, k \in q_\ell\}$ is also a rank-3 tensor. We begin by writing

$$\begin{aligned} K_t^{(2)}(\boldsymbol{x}_\alpha, \boldsymbol{x}_\beta) &= \langle \nabla f(\boldsymbol{x}_\alpha, \boldsymbol{w}_t), \nabla f(\boldsymbol{x}_\beta, \boldsymbol{w}_t) \rangle \\ &= \sum_{\ell \in [L]} \sum_{j \in [p_\ell]} \sum_{k \in [q_\ell]} \frac{\partial f(\boldsymbol{x}_\alpha, \boldsymbol{w}_t)}{\partial \mathbf{W}_{jk}^{(\ell)}} \cdot \frac{\partial f(\boldsymbol{x}_\beta, \boldsymbol{w}_t)}{\partial \mathbf{W}_{jk}^{(\ell)}} \\ &= (\nabla f_\alpha)^{\ell jk} (\nabla f_\beta)_{\ell jk}, \end{aligned}$$

where in the last line we have used the *Einstein notation* [9]. Taking derivative w.r.t. $t$ gives

$$\frac{d}{dt} K_t^{(2)}(\boldsymbol{x}_\alpha, \boldsymbol{x}_\beta) = (\frac{d}{dt} \nabla f_\alpha)^{\ell jk} (\nabla f_\beta)_{\ell jk} + (\nabla f_\alpha)^{\ell jk} (\frac{d}{dt} \nabla f_\beta)_{\ell jk}.$$

We have

$$\begin{aligned} (\frac{d}{dt} \nabla f_\alpha)_{\ell jk} &= \frac{\partial (\nabla f_\alpha)_{\ell jk}}{\partial \mathbf{W}_{suv}} \cdot \frac{d\mathbf{W}_{suv}}{dt} \\ &= \frac{\partial (\nabla f_\alpha)_{\ell jk}}{\partial \mathbf{W}_{suv}} \cdot \frac{-dL(\boldsymbol{w}_t)}{d\mathbf{W}_{suv}} \\ &= \frac{\partial (\nabla f_\alpha)_{\ell jk}}{\partial \mathbf{W}_{suv}} \cdot \bigg( -\frac{1}{n} \sum_{\gamma \in [n]} (f_\gamma - y_\gamma) \cdot (\nabla f_\gamma)_{suv} \bigg) \\ &= -\frac{1}{n} \sum_{\gamma \in [n]} (f_\gamma - y_\gamma) \cdot \frac{\partial (\nabla f_\alpha)_{\ell jk}}{\partial \mathbf{W}_{suv}} \cdot (\nabla f_\gamma)_{suv} \\ &= -\frac{1}{n} \sum_{\gamma \in [n]} (f_\gamma - y_\gamma) \cdot (\nabla^2 f_\alpha)_{suv}^{\ell jk} \cdot (\nabla f_\gamma)_{suv}, \end{aligned}$$

where we denote $\nabla^2 f$ to be the rank-6 tensor, whose $(\ell, j, k, s, u, v)$-th entry is given by

$$\partial (\nabla f)_{\ell jk} / \partial \mathbf{W}_{suv} = \frac{\partial^2 f}{\partial \mathbf{W}_{suv} \mathbf{W}_{\ell jk}}.$$

A similar computation gives

$$(\frac{d}{dt} \nabla f_\beta)_{\ell jk} = -\frac{1}{n} \sum_{\gamma \in [n]} (f_\gamma - y_\gamma) \cdot (\nabla^2 f_\beta)_{suv}^{\ell jk} \cdot (\nabla f_\gamma)_{suv}.$$

Hence, we arrive at

$$
K_t^{(3)}(\boldsymbol{x}_\alpha, \boldsymbol{x}_\beta, \boldsymbol{x}_\gamma)
$$
$$
= (\nabla^2 f_\alpha)_{suv}^{\ell jk}(\nabla f_\beta)_{\ell jk}(\nabla f_\gamma)_{suv} + (\nabla f_\alpha)_{\ell jk}(\nabla^2 f_\beta)_{suv}^{\ell jk}(\nabla f_\gamma)_{suv}
$$
$$
= \sum_{\ell \in [L]} \sum_{j \in [p_\ell]} \sum_{k \in [q_\ell]} \sum_{s \in [L]} \sum_{u \in [p_s]} \sum_{v \in [q_s]} \frac{\partial f(\boldsymbol{x}_\beta, \boldsymbol{w}_t)}{\partial \mathbf{W}_{\ell jk}} \cdot \frac{\partial^2 f(\boldsymbol{x}_\alpha, \boldsymbol{w}_t)}{\partial \mathbf{W}_{\ell jk} \partial \mathbf{W}_{suv}} \cdot \frac{\partial f(\boldsymbol{x}_\gamma, \boldsymbol{w}_t)}{\partial \mathbf{W}_{suv}}
$$
$$
+ \frac{\partial f(\boldsymbol{x}_\alpha, \boldsymbol{w}_t)}{\partial \mathbf{W}_{\ell jk}} \cdot \frac{\partial^2 f(\boldsymbol{x}_\beta, \boldsymbol{w}_t)}{\partial \mathbf{W}_{\ell jk} \partial \mathbf{W}_{suv}} \cdot \frac{\partial f(\boldsymbol{x}_\gamma, \boldsymbol{w}_t)}{\partial \mathbf{W}_{suv}}.
$$

Note that the above expression is a quadratic form:

$$
K_t^{(3)}(\boldsymbol{x}_\alpha, \boldsymbol{x}_\beta, \boldsymbol{x}_\gamma) = (\nabla f_\beta)^\top (\nabla^2 f_\alpha)(\nabla f_\gamma) + (\nabla f_\alpha)^\top (\nabla^2 f_\beta)(\nabla f_\gamma).
$$

We have already seen some patterns showing up. To obtain $K_t^{(3)}$ from $K_t^{(2)}$, we simply conduct the following program:

1. Start with $K_t^{(2)} = (\nabla f_\alpha)^{\ell jk}(\nabla f_\beta)_{\ell jk}$ in Einstein notation, which is a function of $r = 2$ gradients;

2. Introduce a new index $\gamma$ for data points, and a new set of indices $(suv)$ for weights;

3. Replicate $K_t^{(2)}$ for $r = 2$ times, and append $(\nabla f_\gamma)_{suv}$ to the end of each term:

$$
(\nabla f_\alpha)^{\ell jk}(\nabla f_\beta)_{\ell jk}(\nabla f_\gamma)_{suv} + (\nabla f_\alpha)^{\ell jk}(\nabla f_\beta)_{\ell jk}(\nabla f_\gamma)_{suv};
$$

4. Choose a term from a total of $r = 2$ terms in $K_t^{(2)}$, raise its gradient to one higher level, add the new indices $(suv)$ to this term, and do this operation in all possible ways:

$$
K_t^{(3)} = (\nabla^2 f_\alpha)_{suv}^{\ell jk}(\nabla f_\beta)_{\ell jk}(\nabla f_\gamma)_{suv} + (\nabla f_\alpha)_{\ell jk}(\nabla^2 f_\beta)_{suv}^{\ell jk}(\nabla f_\gamma)_{suv}.
$$

We now apply the above program to obtain $K_t^{(4)}$ from $K_t^{(3)}$:

1. Introduce a new index $\xi$ for data points, and a new set of indices $(abc)$;

2. Since $K^{(3)}$ is a function of $r = 3$ gradients, we replicate $K_t^{(3)}$ for 3 times, and append $(\nabla f_\xi)_{abc}$ to the end of each term:

$$
(\nabla^2 f_\alpha)_{suv}^{\ell jk}(\nabla f_\beta)_{\ell jk}(\nabla f_\gamma)_{suv}(\nabla f_\xi)_{abc}
$$
$$
+ (\nabla^2 f_\alpha)_{suv}^{\ell jk}(\nabla f_\beta)_{\ell jk}(\nabla f_\gamma)_{suv}(\nabla f_\xi)_{abc}
$$
$$
+ (\nabla^2 f_\alpha)_{suv}^{\ell jk}(\nabla f_\beta)_{\ell jk}(\nabla f_\gamma)_{suv}(\nabla f_\xi)_{abc}
$$
$$
+ (\nabla f_\alpha)_{\ell jk}(\nabla^2 f_\beta)_{suv}^{\ell jk}(\nabla f_\gamma)_{suv}(\nabla f_\xi)_{abc}
$$
$$
+ (\nabla f_\alpha)_{\ell jk}(\nabla^2 f_\beta)_{suv}^{\ell jk}(\nabla f_\gamma)_{suv}(\nabla f_\xi)_{abc}
$$
$$
+ (\nabla f_\alpha)_{\ell jk}(\nabla^2 f_\beta)_{suv}^{\ell jk}(\nabla f_\gamma)_{suv}(\nabla f_\xi)_{abc};
$$

3. Raise a term's gradient (from the former 3 terms) to a higher level, add the new indices, and do it in all possible ways:

$$
\begin{aligned}
K_t^{(4)} =& (\nabla^3 f_\alpha)_{suv,abc}^{\ell jk}(\nabla f_\beta)_{\ell jk}(\nabla f_\gamma)_{suv}(\nabla f_\xi)_{abc} \\
&+ (\nabla^2 f_\alpha)_{suv}^{\ell jk}(\nabla^2 f_\beta)_{abc}^{\ell jk}(\nabla f_\gamma)_{suv}(\nabla f_\xi)_{abc} \\
&+ (\nabla^2 f_\alpha)_{suv}^{\ell jk}(\nabla f_\beta)_{\ell jk}(\nabla^2 f_\gamma)_{abc}^{suv}(\nabla f_\xi)_{abc} \\
&+ (\nabla^2 f_\alpha)_{abc}^{\ell jk}(\nabla^2 f_\beta)_{suv}^{\ell jk}(\nabla f_\gamma)_{suv}(\nabla f_\xi)_{abc} \\
&+ (\nabla f_\alpha)_{\ell jk}(\nabla^3 f_\beta)_{suv,abc}^{\ell jk}(\nabla f_\gamma)_{suv}(\nabla f_\xi)_{abc} \\
&+ (\nabla f_\alpha)_{\ell jk}(\nabla^2 f_\beta)_{suv}^{\ell jk}(\nabla^2 f_\gamma)_{abc}^{suv}(\nabla f_\xi)_{abc}.
\end{aligned} \tag{11}
$$

In the above program, we have regarded $\nabla^3 f$ as a rank-9 tensor, whose $(\ell, j, k, s, u, v, a, b, c)$-th entry is given by

$$\frac{\partial^3 f}{\partial \mathbf{W}_{abc} \partial \mathbf{W}_{suv} \partial \mathbf{W}_{\ell j k}}.$$

The correctness of the above recursive symbolic program can be proved by straightforward induction, and we omit the details.

## B.2 Explicit Expressions for NTH for Two-Layer Nets

Taking advantage of the recursive program, it's relatively easy to get explicit expressions for $K^{(r)}$ when $r$ is not too large. We focus on the following two-layer net

$$f(\boldsymbol{x}, \boldsymbol{w}) = \frac{1}{\sqrt{m}} \mathbf{a}^\top \sigma(\mathbf{W}\boldsymbol{x}). \tag{12}$$

**Proposition B.1** (Expression for $K_t^{(2)}$, $K_t^{(3)}$ and $K_t^{(4)}$ in a two-layer net). *For the two layer neural network $f(\boldsymbol{x}, \boldsymbol{w}) = \frac{1}{\sqrt{m}} \mathbf{a}^\top \sigma(\mathbf{W}\boldsymbol{x})$, where $\mathbf{a} \in \mathbb{R}^m, \mathbf{W} \in \mathbb{R}^{m \times d}$, and $\sigma$ an activation function, we have*

$$K_t^{(2)}(\boldsymbol{x}_\alpha, \boldsymbol{x}_\beta)$$
$$= \frac{1}{m} \boldsymbol{x}_\alpha^\top \boldsymbol{x}_\beta \cdot \left\langle \sigma'(\mathbf{W}\boldsymbol{x}_\alpha) \odot \mathbf{a}_t, \sigma'(\mathbf{W}\boldsymbol{x}_\beta) \odot \mathbf{a}_t \right\rangle + \frac{1}{m} \left\langle \sigma(\mathbf{W}\boldsymbol{x}_\alpha), \sigma(\mathbf{W}\boldsymbol{x}_\beta) \right\rangle, \tag{13}$$

$$K_t^{(3)}(\boldsymbol{x}_\alpha, \boldsymbol{x}_\beta, \boldsymbol{x}_\gamma)$$
$$= \frac{1}{m\sqrt{m}} \boldsymbol{x}_\alpha^\top \boldsymbol{x}_\gamma \cdot \boldsymbol{x}_\alpha^\top \boldsymbol{x}_\beta \cdot \left\langle \mathbf{a}_t \odot \mathbf{a}_t \odot \mathbf{a}_t, \sigma''(\mathbf{W}\boldsymbol{x}_\alpha) \odot \sigma'(\mathbf{W}\boldsymbol{x}_\beta) \odot \sigma'(\mathbf{W}\boldsymbol{x}_\gamma) \right\rangle$$
$$+ \frac{1}{m\sqrt{m}} \boldsymbol{x}_\beta^\top \boldsymbol{x}_\gamma \cdot \boldsymbol{x}_\alpha^\top \boldsymbol{x}_\beta \cdot \left\langle \mathbf{a}_t \odot \mathbf{a}_t \odot \mathbf{a}_t, \sigma'(\mathbf{W}\boldsymbol{x}_\alpha) \odot \sigma''(\mathbf{W}\boldsymbol{x}_\beta) \odot \sigma'(\mathbf{W}\boldsymbol{x}_\gamma) \right\rangle$$
$$+ \frac{2}{m\sqrt{m}} \boldsymbol{x}_\alpha^\top \boldsymbol{x}_\beta \cdot \left\langle \mathbf{a}_t, \sigma'(\mathbf{W}\boldsymbol{x}_\alpha) \odot \sigma'(\mathbf{W}\boldsymbol{x}_\beta) \odot \sigma(\mathbf{W}\boldsymbol{x}_\gamma) \right\rangle$$
$$+ \frac{1}{m\sqrt{m}} \boldsymbol{x}_\alpha^\top \boldsymbol{x}_\gamma \cdot \left\langle \mathbf{a}_t, \sigma'(\mathbf{W}\boldsymbol{x}_\alpha) \odot \sigma(\mathbf{W}\boldsymbol{x}_\beta) \odot \sigma'(\mathbf{W}\boldsymbol{x}_\gamma) \right\rangle$$
$$+ \frac{1}{m\sqrt{m}} \boldsymbol{x}_\beta^\top \boldsymbol{x}_\gamma \cdot \left\langle \mathbf{a}_t, \sigma(\mathbf{W}\boldsymbol{x}_\alpha) \odot \sigma'(\mathbf{W}\boldsymbol{x}_\beta) \odot \sigma'(\mathbf{W}\boldsymbol{x}_\gamma) \right\rangle, \tag{14}$$

$$K_t^{(4)}(\boldsymbol{x}_\alpha, \boldsymbol{x}_\beta, \boldsymbol{x}_\gamma, \boldsymbol{x}_\xi)$$
$$= \frac{1}{m^2} \boldsymbol{x}_\alpha^\top \boldsymbol{x}_\beta \cdot \boldsymbol{x}_\alpha^\top \boldsymbol{x}_\gamma \cdot \boldsymbol{x}_\alpha^\top \boldsymbol{x}_\xi \cdot \left\langle \mathbf{a}_t^{\odot 4}, \sigma'''(\mathbf{W}\boldsymbol{x}_\alpha) \odot \sigma'(\mathbf{W}\boldsymbol{x}_\beta) \odot \sigma'(\mathbf{W}\boldsymbol{x}_\gamma) \odot \sigma'(\mathbf{W}\boldsymbol{x}_\xi) \right\rangle$$
$$+ \frac{1}{m^2} \boldsymbol{x}_\beta^\top \boldsymbol{x}_\alpha \cdot \boldsymbol{x}_\beta^\top \boldsymbol{x}_\gamma \cdot \boldsymbol{x}_\beta^\top \boldsymbol{x}_\xi \cdot \left\langle \mathbf{a}_t^{\odot 4}, \sigma'(\mathbf{W}\boldsymbol{x}_\alpha) \odot \sigma'''(\mathbf{W}\boldsymbol{x}_\beta) \odot \sigma'(\mathbf{W}\boldsymbol{x}_\gamma) \odot \sigma'(\mathbf{W}\boldsymbol{x}_\xi) \right\rangle$$
$$+ \frac{1}{m^2} \boldsymbol{x}_\alpha^\top \boldsymbol{x}_\beta \cdot (\boldsymbol{x}_\alpha^\top \boldsymbol{x}_\gamma \cdot \boldsymbol{x}_\beta^\top \boldsymbol{x}_\xi + \boldsymbol{x}_\alpha^\top \boldsymbol{x}_\xi \cdot \boldsymbol{x}_\beta^\top \boldsymbol{x}_\gamma) \cdot \left\langle \mathbf{a}^{\odot 4}, \sigma''(\mathbf{W}\boldsymbol{x}_\alpha) \odot \sigma''(\mathbf{W}\boldsymbol{x}_\beta) \odot \sigma'(\mathbf{W}\boldsymbol{x}_\gamma) \odot \sigma'(\mathbf{W}\boldsymbol{x}_\xi) \right\rangle$$
$$+ \frac{1}{m^2} \boldsymbol{x}_\alpha^\top \boldsymbol{x}_\gamma \cdot \boldsymbol{x}_\alpha^\top \boldsymbol{x}_\beta \cdot \boldsymbol{x}_\gamma^\top \boldsymbol{x}_\xi \cdot \left\langle \mathbf{a}^{\odot 4}, \sigma''(\mathbf{W}\boldsymbol{x}_\alpha) \odot \sigma'(\mathbf{W}\boldsymbol{x}_\beta) \odot \sigma''(\mathbf{W}\boldsymbol{x}_\gamma) \odot \sigma'(\mathbf{W}\boldsymbol{x}_\xi) \right\rangle$$
$$+ \frac{1}{m^2} \boldsymbol{x}_\beta^\top \boldsymbol{x}_\gamma \cdot \boldsymbol{x}_\beta^\top \boldsymbol{x}_\alpha \cdot \boldsymbol{x}_\gamma^\top \boldsymbol{x}_\xi \cdot \left\langle \mathbf{a}^{\odot 4}, \sigma'(\mathbf{W}\boldsymbol{x}_\alpha) \odot \sigma''(\mathbf{W}\boldsymbol{x}_\beta) \odot \sigma''(\mathbf{W}\boldsymbol{x}_\gamma) \odot \sigma'(\mathbf{W}\boldsymbol{x}_\xi) \right\rangle$$
$$+ \frac{3}{m^2} \boldsymbol{x}_\alpha^\top \boldsymbol{x}_\beta \cdot \boldsymbol{x}_\alpha^\top \boldsymbol{x}_\gamma \cdot \left\langle \mathbf{a}^{\odot 2}, \sigma''(\mathbf{W}\boldsymbol{x}_\alpha) \odot \sigma'(\mathbf{W}\boldsymbol{x}_\beta) \odot \sigma'(\mathbf{W}\boldsymbol{x}_\gamma) \odot \sigma(\mathbf{W}\boldsymbol{x}_\xi) \right\rangle$$
$$+ \frac{3}{m^2} \boldsymbol{x}_\beta^\top \boldsymbol{x}_\alpha \cdot \boldsymbol{x}_\beta^\top \boldsymbol{x}_\gamma \cdot \left\langle \mathbf{a}^{\odot 2}, \sigma'(\mathbf{W}\boldsymbol{x}_\alpha) \odot \sigma''(\mathbf{W}\boldsymbol{x}_\beta) \odot \sigma'(\mathbf{W}\boldsymbol{x}_\gamma) \odot \sigma(\mathbf{W}\boldsymbol{x}_\xi) \right\rangle$$
$$+ \frac{2}{m^2} \boldsymbol{x}_\alpha^\top \boldsymbol{x}_\beta \cdot \boldsymbol{x}_\alpha^\top \boldsymbol{x}_\xi \cdot \left\langle \mathbf{a}^{\odot 2}, \sigma''(\mathbf{W}\boldsymbol{x}_\alpha) \odot \sigma'(\mathbf{W}\boldsymbol{x}_\beta) \odot \sigma(\mathbf{W}\boldsymbol{x}_\gamma) \odot \sigma'(\mathbf{W}\boldsymbol{x}_\xi) \right\rangle$$

$$+ \frac{2}{m^2} \boldsymbol{x}_\beta^\top \boldsymbol{x}_\alpha \cdot \boldsymbol{x}_\beta^\top \boldsymbol{x}_\xi \cdot \left\langle \mathbf{a}^{\odot 2}, \sigma'(\mathbf{W}\boldsymbol{x}_\alpha) \odot \sigma''(\mathbf{W}\boldsymbol{x}_\beta) \odot \sigma(\mathbf{W}\boldsymbol{x}_\gamma) \odot \sigma'(\mathbf{W}\boldsymbol{x}_\xi) \right\rangle$$

$$+ \frac{2}{m^2} \boldsymbol{x}_\alpha^\top \boldsymbol{x}_\beta \cdot \boldsymbol{x}_\gamma^\top \boldsymbol{x}_\xi \cdot \left\langle \mathbf{a}^{\odot 2}, \sigma'(\mathbf{W}\boldsymbol{x}_\alpha) \odot \sigma'(\mathbf{W}\boldsymbol{x}_\beta) \odot \sigma'(\mathbf{W}\boldsymbol{x}_\gamma) \odot \sigma'(\mathbf{W}\boldsymbol{x}_\xi) \right\rangle$$

$$+ \frac{1}{m^2} \boldsymbol{x}_\alpha^\top \boldsymbol{x}_\gamma \cdot \boldsymbol{x}_\alpha^\top \boldsymbol{x}_\xi \cdot \left\langle \mathbf{a}^{\odot 2}, \sigma''(\mathbf{W}\boldsymbol{x}_\alpha) \odot \sigma(\mathbf{W}\boldsymbol{x}_\beta) \odot \sigma'(\mathbf{W}\boldsymbol{x}_\gamma) \odot \sigma'(\mathbf{W}\boldsymbol{x}_\xi) \right\rangle$$

$$+ \frac{1}{m^2} \boldsymbol{x}_\gamma^\top \boldsymbol{x}_\alpha \cdot \boldsymbol{x}_\gamma^\top \boldsymbol{x}_\xi \cdot \left\langle \mathbf{a}^{\odot 2}, \sigma'(\mathbf{W}\boldsymbol{x}_\alpha) \odot \sigma(\mathbf{W}\boldsymbol{x}_\beta) \odot \sigma''(\mathbf{W}\boldsymbol{x}_\gamma) \odot \sigma'(\mathbf{W}\boldsymbol{x}_\xi) \right\rangle$$

$$+ \frac{1}{m^2} \boldsymbol{x}_\beta^\top \boldsymbol{x}_\gamma \cdot \boldsymbol{x}_\beta^\top \boldsymbol{x}_\xi \cdot \left\langle \mathbf{a}^{\odot 2}, \sigma(\mathbf{W}\boldsymbol{x}_\alpha) \odot \sigma''(\mathbf{W}\boldsymbol{x}_\beta) \odot \sigma'(\mathbf{W}\boldsymbol{x}_\gamma) \odot \sigma'(\mathbf{W}\boldsymbol{x}_\xi) \right\rangle$$

$$+ \frac{1}{m^2} \boldsymbol{x}_\gamma^\top \boldsymbol{x}_\beta \cdot \boldsymbol{x}_\gamma^\top \boldsymbol{x}_\xi \cdot \left\langle \mathbf{a}^{\odot 2}, \sigma(\mathbf{W}\boldsymbol{x}_\alpha) \odot \sigma'(\mathbf{W}\boldsymbol{x}_\beta) \odot \sigma''(\mathbf{W}\boldsymbol{x}_\gamma) \odot \sigma'(\mathbf{W}\boldsymbol{x}_\xi) \right\rangle$$

$$+ \frac{1}{m^2} (\boldsymbol{x}_\alpha^\top \boldsymbol{x}_\gamma \cdot \boldsymbol{x}_\beta^\top \boldsymbol{x}_\xi + \boldsymbol{x}_\alpha^\top \boldsymbol{x}_\xi \cdot \boldsymbol{x}_\beta^\top \boldsymbol{x}_\gamma) \cdot \left\langle \mathbf{a}^{\odot 2}, \sigma'(\mathbf{W}\boldsymbol{x}_\alpha) \odot \sigma'(\mathbf{W}\boldsymbol{x}_\beta) \odot \sigma'(\mathbf{W}\boldsymbol{x}_\gamma) \odot \sigma'(\mathbf{W}\boldsymbol{x}_\xi) \right\rangle$$

$$+ \frac{2}{m^2} \boldsymbol{x}_\alpha^\top \boldsymbol{x}_\beta \cdot \left\langle \mathbf{1}_m, \sigma'(\mathbf{W}\boldsymbol{x}_\alpha) \odot \sigma'(\mathbf{W}\boldsymbol{x}_\beta) \odot \sigma(\mathbf{W}\boldsymbol{x}_\gamma) \odot \sigma(\mathbf{W}\boldsymbol{x}_\xi) \right\rangle$$

$$+ \frac{1}{m^2} \boldsymbol{x}_\alpha^\top \boldsymbol{x}_\gamma \cdot \left\langle \mathbf{1}_m, \sigma'(\mathbf{W}\boldsymbol{x}_\alpha) \odot \sigma(\mathbf{W}\boldsymbol{x}_\beta) \odot \sigma'(\mathbf{W}\boldsymbol{x}_\gamma) \odot \sigma(\mathbf{W}\boldsymbol{x}_\xi) \right\rangle$$

$$+ \frac{1}{m^2} \boldsymbol{x}_\beta^\top \boldsymbol{x}_\gamma \cdot \left\langle \mathbf{1}_m, \sigma(\mathbf{W}\boldsymbol{x}_\alpha) \odot \sigma'(\mathbf{W}\boldsymbol{x}_\beta) \odot \sigma'(\mathbf{W}\boldsymbol{x}_\gamma) \odot \sigma(\mathbf{W}\boldsymbol{x}_\xi) \right\rangle, \tag{15}$$

*where we let* $\mathbf{a}^{\odot r}$ *to be the vector whose* $i$*-th entry is* $\mathbf{a}_i^r$*.*

*Proof.* **Computation of** $K_t^{(2)}$**.** We first compute $K_t^{(2)}(\boldsymbol{x}_\alpha, \boldsymbol{x}_\beta) = (\nabla f_\alpha)_{\ell jk} (\nabla f_\beta)_{\ell jk}$. We have

$$\frac{\partial f(\boldsymbol{x}, \boldsymbol{w})}{\partial \mathbf{a}} = \frac{1}{\sqrt{m}} \sigma(\mathbf{W}\boldsymbol{x})$$

$$\frac{\partial f(\boldsymbol{x}, \boldsymbol{w})}{\partial \mathbf{W}} = \frac{1}{\sqrt{m}} \operatorname{diag}(\sigma'(\mathbf{W}\boldsymbol{x})\mathbf{a}\boldsymbol{x}^\top = \frac{1}{\sqrt{m}} \left( \sigma'(\mathbf{W}\boldsymbol{x} \odot \mathbf{a}) \right) x^\top.$$

Then Equation (13) follows by trivial algebra.

**Computation of** $K_t^{(3)}$**.** Now consider $K_t^{(3)}$. We have

$$K_t^{(3)}(\boldsymbol{x}_\alpha, \boldsymbol{x}_\beta, \boldsymbol{x}_\gamma) = (\nabla^2 f_\alpha)_{suv}^{\ell jk} (\nabla f_\beta)_{\ell jk} (\nabla f_\gamma)_{suv} + (\nabla f_\alpha)_{\ell jk} (\nabla^2 f_\beta)_{suv}^{\ell jk} (\nabla f_\gamma)_{suv}.$$

For the $\ell = s = 2$ term, we have $(\nabla^2 f)_{2uv}^{2jk} = 0$, since $\partial f / \partial \mathbf{a}$ is constant in $\mathbf{a}$. For the $\ell = s = 1$ term, we have

$$(\nabla^2 f)_{1uv}^{1jk} = \frac{\partial^2 f(\boldsymbol{x}, \boldsymbol{w})}{\partial \mathbf{W}_{uv} W_{jk}}$$

$$= \frac{\partial}{\partial \mathbf{W}_{uv}} \frac{1}{\sqrt{m}} \sigma'(\mathbf{W}\boldsymbol{x}_j)\mathbf{a}_j \boldsymbol{x}_k$$

$$= \frac{1}{\sqrt{m}} \sigma''(\mathbf{W}\boldsymbol{x}_j)\delta_{ju} \boldsymbol{x}_v \mathbf{a}_j \boldsymbol{x}_k,$$

where $\delta_{ju}$ is the Kronecker delta function. Hence we have

$$(\nabla^2 f_\alpha)_{1uv}^{1jk} (\nabla f_\beta)_{1jk} (\nabla f_\gamma)_{1uv} = \frac{1}{m\sqrt{m}} \sigma''(\mathbf{W}\boldsymbol{x}_\alpha)_j \sigma'(\mathbf{W}\boldsymbol{x}_\beta)_j \sigma'(\mathbf{W}\boldsymbol{x}_\gamma)_j \mathbf{a}_j^3 \boldsymbol{x}_{\alpha,v} \boldsymbol{x}_{\gamma,v} \boldsymbol{x}_{\alpha,k} \boldsymbol{x}_{\beta,k}$$

$$= \frac{1}{m\sqrt{m}} \boldsymbol{x}_\alpha^\top \boldsymbol{x}_\gamma \cdot \boldsymbol{x}_\alpha^\top \boldsymbol{x}_\beta \cdot \left\langle \mathbf{a} \odot \mathbf{a} \odot \mathbf{a}, \sigma''(\mathbf{W}\boldsymbol{x}_\alpha) \odot \sigma'(\mathbf{W}\boldsymbol{x}_\beta) \odot \sigma'(\mathbf{W}\boldsymbol{x}_\gamma) \right\rangle.$$

A similar computation gives

$$(\nabla f_\alpha)_{1jk} (\nabla^2 f_\beta)_{1uv}^{1jk} (\nabla f_\gamma)_{1uv} = \frac{1}{m\sqrt{m}} \boldsymbol{x}_\beta^\top \boldsymbol{x}_\gamma \cdot \boldsymbol{x}_\alpha^\top \boldsymbol{x}_\beta \cdot \left\langle \mathbf{a} \odot \mathbf{a} \odot \mathbf{a}, \sigma'(\mathbf{W}\boldsymbol{x}_\alpha) \odot \sigma''(\mathbf{W}\boldsymbol{x}_\beta) \odot \sigma'(\mathbf{W}\boldsymbol{x}_\gamma) \right\rangle.$$

For the $\ell = 1, s = 2$ term, note that $a$ is a vector (also a row matrix), so $u = 1$. This gives

$$(\nabla^2 f_\alpha)^{1jk}_{21v}(\nabla f_\beta)_{1jk}(\nabla f_\gamma)_{21v} = \frac{\partial}{\partial \mathbf{a}_v}\left(\frac{1}{\sqrt{m}}\sigma'(\mathbf{W}\boldsymbol{x}_\alpha)_j\mathbf{a}_j\boldsymbol{x}_{\alpha,k}\right)\cdot\frac{1}{\sqrt{m}}\sigma'(\mathbf{W}\boldsymbol{x}_\beta)_j\mathbf{a}_j\boldsymbol{x}_{\beta,k}\cdot\frac{1}{\sqrt{m}}\sigma(\mathbf{W}\boldsymbol{x}_\gamma)_v$$

$$= \frac{1}{m\sqrt{m}}\sigma'(\mathbf{W}\boldsymbol{x}_\alpha)_j\delta_{jv}\boldsymbol{x}_{\alpha,k}\cdot\sigma'(\mathbf{W}\boldsymbol{x}_\beta)_j\mathbf{a}_j\boldsymbol{x}_{\beta,k}\cdot\sigma(\mathbf{W}\boldsymbol{x}_\gamma)_v$$

$$= \frac{1}{m\sqrt{m}}\sigma'(\mathbf{W}\boldsymbol{x}_\alpha)_j\sigma'(\mathbf{W}\boldsymbol{x}_\beta)_j\sigma(\mathbf{W}\boldsymbol{x}_\gamma)_j\mathbf{a}_j\boldsymbol{x}_{\alpha,k}\boldsymbol{x}_{\beta,k}$$

$$= \frac{1}{m\sqrt{m}}\boldsymbol{x}_\alpha^\top\boldsymbol{x}_\beta\cdot\Big\langle a, \sigma'(\mathbf{W}\boldsymbol{x}_\alpha)\odot\sigma'(\mathbf{W}\boldsymbol{x}_\beta)\odot\sigma(\mathbf{W}\boldsymbol{x}_\gamma)\Big\rangle.$$

By symmetry, the term $(\nabla f_\alpha)_{1jk}(\nabla^2 f_\beta)^{1jk}_{21v}(\nabla f_\gamma)_{21v}$ is also equal to the above quantity. Finally, we calculate the $\ell = 2, s = 1$ term. Note that in this case, $j = 1$. Hence, we have

$$(\nabla^2 f_\alpha)^{21k}_{1uv}(\nabla f_\beta)_{21k}(\nabla f_\gamma)_{1uv} = \frac{\partial}{\partial \mathbf{W}_{uv}}\left(\frac{1}{\sqrt{m}}\sigma(\mathbf{W}\boldsymbol{x}_\alpha)_k\right)\cdot\frac{1}{\sqrt{m}}\sigma(\mathbf{W}\boldsymbol{x}_\beta)_k\cdot\frac{1}{\sqrt{m}}\sigma'(\mathbf{W}\boldsymbol{x}_\gamma)_u\mathbf{a}_u\boldsymbol{x}_{\gamma,v}$$

$$= \frac{1}{m\sqrt{m}}\sigma'(\mathbf{W}\boldsymbol{x}_\alpha)_k\delta_{uk}\boldsymbol{x}_{\alpha,v}\cdot\sigma(\mathbf{W}\boldsymbol{x}_\beta)_k\cdot\sigma'(\mathbf{W}\boldsymbol{x}_\gamma)_u\mathbf{a}_u\boldsymbol{x}_{\gamma,v}$$

$$= \frac{1}{m\sqrt{m}}\sigma'(\mathbf{W}\boldsymbol{x}_\alpha)_k\sigma(\mathbf{W}\boldsymbol{x}_\beta)_k\sigma'(\mathbf{W}\boldsymbol{x}_\gamma)_k\mathbf{a}_k\boldsymbol{x}_{\alpha,v}\boldsymbol{x}_{\gamma,v}$$

$$= \frac{1}{m\sqrt{m}}\boldsymbol{x}_\alpha^\top\boldsymbol{x}_\gamma\cdot\Big\langle a, \sigma'(\mathbf{W}\boldsymbol{x}_\alpha)\odot\sigma(\mathbf{W}\boldsymbol{x}_\beta)\odot\sigma'(\mathbf{W}\boldsymbol{x}_\gamma)\Big\rangle.$$

A similar computation gives

$$(\nabla f_\alpha)_{21k}(\nabla^2 f_\beta)^{21k}_{1uv}(\nabla f_\gamma)_{1uv} = \frac{1}{m\sqrt{m}}\boldsymbol{x}_\beta^\top\boldsymbol{x}_\gamma\cdot\Big\langle a, \sigma(\mathbf{W}\boldsymbol{x}_\alpha)\odot\sigma'(\mathbf{W}\boldsymbol{x}_\beta)\odot\sigma'(\mathbf{W}\boldsymbol{x}_\gamma)\Big\rangle.$$

Combining above terms proves Equation (14).

**Computation of $K_4^{(4)}$.** Recall that

$$K_t^{(4)} = (\nabla^3 f_\alpha)^{\ell jk}_{suv,abc}(\nabla f_\beta)_{\ell jk}(\nabla f_\gamma)_{suv}(\nabla f_\xi)_{abc}$$

$$+ (\nabla^2 f_\alpha)^{\ell jk}_{suv}(\nabla^2 f_\beta)^{\ell jk}_{abc}(\nabla f_\gamma)_{suv}(\nabla f_\xi)_{abc}$$

$$+ (\nabla^2 f_\alpha)^{\ell jk}_{suv}(\nabla f_\beta)_{\ell jk}(\nabla^2 f_\gamma)^{suv}_{abc}(\nabla f_\xi)_{abc}$$

$$+ (\nabla^2 f_\alpha)^{\ell jk}_{abc}(\nabla^2 f_\beta)^{\ell jk}_{suv}(\nabla f_\gamma)_{suv}(\nabla f_\xi)_{abc}$$

$$+ (\nabla f_\alpha)_{\ell jk}(\nabla^3 f_\beta)^{\ell jk}_{suv,abc}(\nabla f_\gamma)_{suv}(\nabla f_\xi)_{abc}$$

$$+ (\nabla f_\alpha)_{\ell jk}(\nabla^2 f_\beta)^{\ell jk}_{suv}(\nabla^2 f_\gamma)^{suv}_{abc}(\nabla f_\xi)_{abc}.$$

We denote the six terms above as I, II, III, IV, V, VI. We first calculate some useful quantities. Recall that

$$(\nabla f)_{21k} = \frac{1}{\sqrt{m}}\sigma(\mathbf{W}\boldsymbol{x}_k), \qquad (\nabla f)_{1jk} = \frac{1}{\sqrt{m}}\sigma'(\mathbf{W}\boldsymbol{x}_j)\mathbf{a}_j\boldsymbol{x}_k.$$

In the calculation for $K_t^{(3)}$, we have shown that

$$(\nabla^2 f)^{21k}_{21v} = 0$$

$$(\nabla^2 f)^{21k}_{1uv} = (\nabla^2 f)^{1uv}_{21k} = \frac{1}{\sqrt{m}}\sigma'(\mathbf{W}\boldsymbol{x}_k)\delta_{uk}\boldsymbol{x}_v$$

$$(\nabla^2 f)^{1jk}_{1uv} = (\nabla^2 f)^{1uv}_{1jk} = \frac{1}{\sqrt{m}}\sigma''(\mathbf{W}\boldsymbol{x}_j)\mathbf{a}_j\delta_{ju}\boldsymbol{x}_k\boldsymbol{x}_v.$$

For the third derivative, we note that the expression $(\nabla^3 f)_{suv,abc}^{\ell jk}$ is invariant to permutations of the three triplets $(\ell jk), (suv), (abc)$. Some algebra gives the following identities:

$$(\nabla^3 f)_{21v,21c}^{21k} = (\nabla^3 f)_{21v,1bc}^{21k} = 0$$

$$(\nabla^3 f)_{1uv,21c}^{1jk} = \frac{1}{\sqrt{m}} \sigma''(\mathbf{W}\boldsymbol{x}_j) \delta_{ju} \delta_{jc} \boldsymbol{x}_k \boldsymbol{x}_v$$

$$(\nabla^3 f)_{1uv,1bc}^{1jk} = \frac{1}{\sqrt{m}} \sigma'''(\mathbf{W}\boldsymbol{x}_j) \mathbf{a}_j \delta_{bj} \delta_{ju} \boldsymbol{x}_k \boldsymbol{x}_v \boldsymbol{x}_c.$$

We now calculate expression for $K_t^{(4)}$ based on different configurations of layer indices $(\ell, s, a) \in \{1, 2\}^3$.

1. If $\ell = s = a = 2$, or if $\ell = s = 2, a = 1$, then all six terms are zero.

2. If $\ell = a = 2, s = 1$, then I = II = IV = V = 0. The third term is

$$\text{III} = (\nabla^2 f_\alpha)_{1uv}^{21k}(\nabla f_\beta)_{21k}(\nabla^2 f_\gamma)_{21c}^{1uv}(\nabla f_\xi)_{21c}$$

$$= \frac{1}{m^2} \sigma'(\mathbf{W}\boldsymbol{x}_\alpha)_k \delta_{uk}\boldsymbol{x}_{\alpha,v} \cdot \sigma(\mathbf{W}\boldsymbol{x}_\beta)_k \cdot \sigma'(\mathbf{W}\boldsymbol{x}_\gamma)_c \delta_{uc}\boldsymbol{x}_{\gamma,v} \cdot \sigma(\mathbf{W}\boldsymbol{x}_\xi)_c$$

$$= \frac{1}{m^2} \boldsymbol{x}_\alpha^\top \boldsymbol{x}_\gamma \Big\langle \mathbf{1}_m, \sigma'(\mathbf{W}\boldsymbol{x}_\alpha) \odot \sigma(\mathbf{W}\boldsymbol{x}_\beta) \odot \sigma'(\mathbf{W}\boldsymbol{x}_\gamma) \odot \sigma(\mathbf{W}\boldsymbol{x}_\xi) \Big\rangle.$$

A similar calculation gives

$$\text{VI} = \frac{1}{m^2} \boldsymbol{x}_\beta^\top \boldsymbol{x}_\gamma \Big\langle \mathbf{1}_m, \sigma(\mathbf{W}\boldsymbol{x}_\alpha) \odot \sigma'(\mathbf{W}\boldsymbol{x}_\beta) \odot \sigma'(\mathbf{W}\boldsymbol{x}_\gamma) \odot \sigma(\mathbf{W}\boldsymbol{x}_\xi) \Big\rangle.$$

3. If $s = a = 2, \ell = 1$, then I = III = V = VI = 0. And we have

$$\text{II} = (\nabla^2 f_\alpha)_{21v}^{1jk}(\nabla^2 f_\beta)_{21c}^{1jk}(\nabla f_\gamma)_{21v}(\nabla f_\xi)_{21c}$$

$$= \frac{1}{m^2} \boldsymbol{x}_\alpha^\top \boldsymbol{x}_\beta \Big\langle \mathbf{1}_m, \sigma'(\mathbf{W}\boldsymbol{x}_\alpha) \odot \sigma'(\mathbf{W}\boldsymbol{x}_\beta) \odot \sigma(\mathbf{W}\boldsymbol{x}_\gamma) \odot \sigma(\mathbf{W}\boldsymbol{x}_\xi) \Big\rangle.$$

A similar calculation shows that IV = II.

4. If $\ell = s = 1, a = 2$, then we have

$$\text{I} = (\nabla^3 f_\alpha)_{1uv,21c}^{1jk}(\nabla f_\beta)_{1jk}(\nabla f_\gamma)_{1uv}(\nabla f_\xi)_{21c}$$

$$= \frac{1}{m^2} \boldsymbol{x}_\alpha^\top \boldsymbol{x}_\gamma \cdot \boldsymbol{x}_\alpha^\top \boldsymbol{x}_\beta \Big\langle \mathbf{a}^{\odot 2}, \sigma''(\mathbf{W}\boldsymbol{x}_\alpha) \odot \sigma'(\mathbf{W}\boldsymbol{x}_\beta) \odot \sigma'(\mathbf{W}\boldsymbol{x}_\gamma) \odot \sigma(\mathbf{W}\boldsymbol{x}_\xi) \Big\rangle.$$

Meanwhile, we have

$$\text{II} = (\nabla^2 f_\alpha)_{1uv}^{1jk}(\nabla^2 f_\beta)_{21c}^{1jk}(\nabla f_\gamma)_{1uv}(\nabla f_\xi)_{21c}$$

$$= \frac{1}{m^2} \boldsymbol{x}_\alpha^\top \boldsymbol{x}_\beta \cdot \boldsymbol{x}_\alpha^\top \boldsymbol{x}_\gamma \Big\langle \mathbf{a}^{\odot 2}, \sigma''(\mathbf{W}\boldsymbol{x}_\alpha) \odot \sigma'(\mathbf{W}\boldsymbol{x}_\beta) \odot \sigma'(\mathbf{W}\boldsymbol{x}_\gamma) \odot \sigma(\mathbf{W}\boldsymbol{x}_\xi) \Big\rangle$$

$$= \text{I}.$$

A similar calculation shows that III = II = I. On the other hand, it's easy to check that

$$\text{IV} = \text{V} = \text{VI} = \frac{1}{m^2} \boldsymbol{x}_\beta^\top \boldsymbol{x}_\alpha \cdot \boldsymbol{x}_\beta^\top \boldsymbol{x}_\gamma \Big\langle \mathbf{a}^{\odot 2}, \sigma'(\mathbf{W}\boldsymbol{x}_\alpha) \odot \sigma''(\mathbf{W}\boldsymbol{x}_\beta) \odot \sigma'(\mathbf{W}\boldsymbol{x}_\gamma) \odot \sigma(\mathbf{W}\boldsymbol{x}_\xi) \Big\rangle.$$

5. If $\ell = a = 1, s = 2$, then one can check that

$$\text{I} = \text{IV} = \frac{1}{m^2} \boldsymbol{x}_\alpha^\top \boldsymbol{x}_\beta \cdot \boldsymbol{x}_\alpha^\top \boldsymbol{x}_\xi \Big\langle \mathbf{a}^{\odot 2}, \sigma''(\mathbf{W}\boldsymbol{x}_\alpha) \odot \sigma'(\mathbf{W}\boldsymbol{x}_\beta) \odot \sigma(\mathbf{W}\boldsymbol{x}_\gamma) \odot \sigma'(\mathbf{W}\boldsymbol{x}_\xi) \Big\rangle,$$

and that

$$\text{II} = \text{V} = \frac{1}{m^2} \boldsymbol{x}_\beta^\top \boldsymbol{x}_\alpha \cdot \boldsymbol{x}_\beta^\top \boldsymbol{x}_\xi \Big\langle \mathbf{a}^{\odot 2}, \sigma'(\mathbf{W}\boldsymbol{x}_\alpha) \odot \sigma''(\mathbf{W}\boldsymbol{x}_\beta) \odot \sigma(\mathbf{W}\boldsymbol{x}_\gamma) \odot \sigma'(\mathbf{W}\boldsymbol{x}_\xi) \Big\rangle,$$

On the other hand, we have

$$\text{III} = \text{VI} \frac{1}{m^2} \boldsymbol{x}_\alpha^\top \boldsymbol{x}_\beta \cdot \boldsymbol{x}_\gamma^\top \boldsymbol{x}_\xi \Big\langle \mathbf{a}^{\odot 2}, \sigma'(\mathbf{W}\boldsymbol{x}_\alpha) \odot \sigma'(\mathbf{W}\boldsymbol{x}_\beta) \odot \sigma'(\mathbf{W}\boldsymbol{x}_\gamma) \odot \sigma'(\mathbf{W}\boldsymbol{x}_\xi) \Big\rangle.$$

6. If $s = a = 1, \ell = 2$, then we have

$$\mathrm{I} = \frac{1}{m^2}\boldsymbol{x}_\alpha^\top\boldsymbol{x}_\gamma \cdot \boldsymbol{x}_\alpha^\top\boldsymbol{x}_\xi \left\langle \mathbf{a}^{\odot 2}, \sigma''(\mathbf{W}\boldsymbol{x}_\alpha) \odot \sigma(\mathbf{W}\boldsymbol{x}_\beta) \odot \sigma'(\mathbf{W}\boldsymbol{x}_\gamma) \odot \sigma'(\mathbf{W}\boldsymbol{x}_\xi) \right\rangle$$

$$\mathrm{II} = \frac{1}{m^2}\boldsymbol{x}_\alpha^\top\boldsymbol{x}_\gamma \cdot \boldsymbol{x}_\beta^\top\boldsymbol{x}_\xi \left\langle \mathbf{a}^{\odot 2}, \sigma'(\mathbf{W}\boldsymbol{x}_\alpha) \odot \sigma'(\mathbf{W}\boldsymbol{x}_\beta) \odot \sigma'(\mathbf{W}\boldsymbol{x}_\gamma) \odot \sigma'(\mathbf{W}\boldsymbol{x}_\xi) \right\rangle$$

$$\mathrm{III} = \frac{1}{m^2}\boldsymbol{x}_\gamma^\top\boldsymbol{x}_\alpha \cdot \boldsymbol{x}_\gamma^\top\boldsymbol{x}_\xi \left\langle \mathbf{a}^{\odot 2}, \sigma'(\mathbf{W}\boldsymbol{x}_\alpha) \odot \sigma(\mathbf{W}\boldsymbol{x}_\beta) \odot \sigma''(\mathbf{W}\boldsymbol{x}_\gamma) \odot \sigma'(\mathbf{W}\boldsymbol{x}_\xi) \right\rangle$$

$$\mathrm{IV} = \frac{1}{m^2}\boldsymbol{x}_\alpha^\top\boldsymbol{x}_\xi \cdot \boldsymbol{x}_\beta^\top\boldsymbol{x}_\gamma \left\langle \mathbf{a}^{\odot 2}, \sigma'(\mathbf{W}\boldsymbol{x}_\alpha) \odot \sigma'(\mathbf{W}\boldsymbol{x}_\beta) \odot \sigma'(\mathbf{W}\boldsymbol{x}_\gamma) \odot \sigma'(\mathbf{W}\boldsymbol{x}_\xi) \right\rangle$$

$$\mathrm{V} = \frac{1}{m^2}\boldsymbol{x}_\beta^\top\boldsymbol{x}_\gamma \cdot \boldsymbol{x}_\beta^\top\boldsymbol{x}_\xi \left\langle \mathbf{a}^{\odot 2}, \sigma(\mathbf{W}\boldsymbol{x}_\alpha) \odot \sigma''(\mathbf{W}\boldsymbol{x}_\beta) \odot \sigma'(\mathbf{W}\boldsymbol{x}_\gamma) \odot \sigma'(\mathbf{W}\boldsymbol{x}_\xi) \right\rangle$$

$$\mathrm{VI} = \frac{1}{m^2}\boldsymbol{x}_\gamma^\top\boldsymbol{x}_\beta \cdot \boldsymbol{x}_\gamma^\top\boldsymbol{x}_\xi \left\langle \mathbf{a}^{\odot 2}, \sigma(\mathbf{W}\boldsymbol{x}_\alpha) \odot \sigma'(\mathbf{W}\boldsymbol{x}_\beta) \odot \sigma''(\mathbf{W}\boldsymbol{x}_\gamma) \odot \sigma'(\mathbf{W}\boldsymbol{x}_\xi) \right\rangle.$$

7. If $\ell = a = s = 1$, then we have

$$\mathrm{I} = (\nabla^3 f_\alpha)_{1uv,1bc}^{1jk}(\nabla f_\beta)_{1jk}(\nabla f_\gamma)_{1uv}(\nabla f_\xi)_{1bc}$$

$$= \frac{1}{m^2}\sigma'''(\mathbf{W}\boldsymbol{x}_\alpha)_j\mathbf{a}_j\delta_{bj}\delta_{ju}\boldsymbol{x}_{\alpha,c}\boldsymbol{x}_{\alpha,v}\boldsymbol{x}_{\alpha,k} \cdot \sigma'(\mathbf{W}\boldsymbol{x}_\beta)_j\mathbf{a}_j\boldsymbol{x}_{\beta,k} \cdot \sigma'(\mathbf{W}\boldsymbol{x}_\gamma)_u\mathbf{a}_u\boldsymbol{x}_{\gamma,v} \cdot \sigma'(\mathbf{W}\boldsymbol{x}_\xi)_b\mathbf{a}_b\boldsymbol{x}_{\xi,c}$$

$$= \frac{1}{m^2}\boldsymbol{x}_\alpha^\top\boldsymbol{x}_\xi \cdot \boldsymbol{x}_\alpha^\top\boldsymbol{x}_\gamma \cdot \boldsymbol{x}_\alpha^\top\boldsymbol{x}_\beta \left\langle \mathbf{a}^{\odot 4}, \sigma'''(\mathbf{W}\boldsymbol{x}_\alpha) \odot \sigma'(\mathbf{W}\boldsymbol{x}_\beta) \odot \sigma'(\mathbf{W}\boldsymbol{x}_\gamma) \odot \sigma'(\mathbf{W}\boldsymbol{x}_\xi) \right\rangle.$$

Meanwhile, we have

$$\mathrm{II} = (\nabla^2 f_\alpha)_{1uv}^{1jk}(\nabla^2 f_\beta)_{1bc}^{1jk}(\nabla f_\gamma)_{1uv}(\nabla f_\xi)_{1bc}$$

$$= \frac{1}{m^2}\sigma''(\mathbf{W}\boldsymbol{x}_\alpha)_j\mathbf{a}_j\delta_{ju}\boldsymbol{x}_{\alpha,k}\boldsymbol{x}_{\alpha,v} \cdot \sigma''(\mathbf{W}\boldsymbol{x}_\beta)_j\mathbf{a}_j\delta_{jb}\boldsymbol{x}_{\beta,k}\boldsymbol{x}_{\beta,c} \cdot \sigma'(\mathbf{W}\boldsymbol{x}_\gamma)_u\mathbf{a}_u\boldsymbol{x}_{\gamma,v} \cdot \sigma'(\mathbf{W}\boldsymbol{x}_\xi)_b\mathbf{a}_b\boldsymbol{x}_{\xi,c}$$

$$= \frac{1}{m^2}\boldsymbol{x}_\alpha^\top\boldsymbol{x}_\beta \cdot \boldsymbol{x}_\alpha^\top\boldsymbol{x}_\gamma \cdot \boldsymbol{x}_\beta^\top\boldsymbol{x}_\xi \left\langle \mathbf{a}^{\odot 4}, \sigma''(\mathbf{W}\boldsymbol{x}_\alpha) \odot \sigma''(\mathbf{W}\boldsymbol{x}_\beta) \odot \sigma'(\mathbf{W}\boldsymbol{x}_\gamma) \odot \sigma'(\mathbf{W}\boldsymbol{x}_\xi) \right\rangle.$$

The other terms are calculated similarly:

$$\mathrm{III} = \frac{1}{m^2}\boldsymbol{x}_\alpha^\top\boldsymbol{x}_\gamma \cdot \boldsymbol{x}_\alpha^\top\boldsymbol{x}_\beta \cdot \boldsymbol{x}_\gamma^\top\boldsymbol{x}_\xi \left\langle \mathbf{a}^{\odot 4}, \sigma''(\mathbf{W}\boldsymbol{x}_\alpha) \odot \sigma'(\mathbf{W}\boldsymbol{x}_\beta) \odot \sigma''(\mathbf{W}\boldsymbol{x}_\gamma) \odot \sigma'(\mathbf{W}\boldsymbol{x}_\xi) \right\rangle$$

$$\mathrm{IV} = \frac{1}{m^2}\boldsymbol{x}_\alpha^\top\boldsymbol{x}_\beta \cdot \boldsymbol{x}_\alpha^\top\boldsymbol{x}_\xi \cdot \boldsymbol{x}_\beta^\top\boldsymbol{x}_\gamma \left\langle \mathbf{a}^{\odot 4}, \sigma''(\mathbf{W}\boldsymbol{x}_\alpha) \odot \sigma''(\mathbf{W}\boldsymbol{x}_\beta) \odot \sigma'(\mathbf{W}\boldsymbol{x}_\gamma) \odot \sigma'(\mathbf{W}\boldsymbol{x}_\xi) \right\rangle$$

$$\mathrm{V} = \frac{1}{m^2}\boldsymbol{x}_\beta^\top\boldsymbol{x}_\alpha \cdot \boldsymbol{x}_\beta^\top\boldsymbol{x}_\gamma \cdot \boldsymbol{x}_\beta^\top\boldsymbol{x}_\xi \left\langle \mathbf{a}^{\odot 4}, \sigma'(\mathbf{W}\boldsymbol{x}_\alpha) \odot \sigma'''(\mathbf{W}\boldsymbol{x}_\beta) \odot \sigma'(\mathbf{W}\boldsymbol{x}_\gamma) \odot \sigma'(\mathbf{W}\boldsymbol{x}_\xi) \right\rangle$$

$$\mathrm{VI} = \frac{1}{m^2}\boldsymbol{x}_\beta^\top\boldsymbol{x}_\gamma \cdot \boldsymbol{x}_\beta^\top\boldsymbol{x}_\alpha \cdot \boldsymbol{x}_\gamma^\top\boldsymbol{x}_\xi \left\langle \mathbf{a}^{\odot 4}, \sigma'(\mathbf{W}\boldsymbol{x}_\alpha) \odot \sigma''(\mathbf{W}\boldsymbol{x}_\beta) \odot \sigma''(\mathbf{W}\boldsymbol{x}_\gamma) \odot \sigma'(\mathbf{W}\boldsymbol{x}_\xi) \right\rangle.$$

Putting the above terms together gives Equation (15). $\qquad\square$

## B.3 Expected Values w.r.t. Gaussian Initialization

We now consider the expected values of $K_t^{(r)}$ at initialization, where both $\mathbf{a}_0$ and $\mathbf{W}_0$ have i.i.d. $\mathcal{N}(0,1)$ entries. We will focus the ReLU activation:

$$\sigma(x) = \max(0, x). \tag{16}$$

Technically, $\sigma(\cdot)$ only has a subdifferential at zero, but since Gaussian initialization puts zero mass at this point, we can safely write $\sigma'(x) = \mathbb{1}\{x \geq 0\}$. Moreover, we have $\sigma''(x) = \delta(x)$, where $\delta(\cdot)$ is the Dirac delta function, and $\sigma'''(x) = \delta'(x)$, where $\delta'(\cdot)$ is the distributional derivative of $\delta(\cdot)$. In

this sense, many terms in $K_t^{(4)}$ are not well-defined, if we don't take expectation. For example, the following terms are not well-defined functions:

$$\left\langle \mathbf{a}_t^{\odot 4}, \sigma'''(\mathbf{W}\boldsymbol{x}_\alpha) \odot \sigma'(\mathbf{W}\boldsymbol{x}_\beta) \odot \sigma'(\mathbf{W}\boldsymbol{x}_\gamma) \odot \sigma'(\mathbf{W}\boldsymbol{x}_\xi) \right\rangle$$

$$\left\langle \mathbf{a}^{\odot 4}, \sigma''(\mathbf{W}\boldsymbol{x}_\alpha) \odot \sigma''(\mathbf{W}\boldsymbol{x}_\beta) \odot \sigma'(\mathbf{W}\boldsymbol{x}_\gamma) \odot \sigma'(\mathbf{W}\boldsymbol{x}_\xi) \right\rangle$$

$$\left\langle \mathbf{a}^{\odot 4}, \sigma''(\mathbf{W}\boldsymbol{x}_\alpha) \odot \sigma(\mathbf{W}\boldsymbol{x}_\beta) \odot \sigma'(\mathbf{W}\boldsymbol{x}_\gamma) \odot \sigma'(\mathbf{W}\boldsymbol{x}_\xi) \right\rangle.$$

So it is necessary to integrate over the Gaussian measure to actually make sense of the above expressions.

On the other hand, the following expressions are well-defined functions:

$$\left\langle \mathbf{a}^{\odot 2}, \sigma'(\mathbf{W}\boldsymbol{x}_\alpha) \odot \sigma'(\mathbf{W}\boldsymbol{x}_\beta) \odot \sigma'(\mathbf{W}\boldsymbol{x}_\gamma) \odot \sigma'(\mathbf{W}\boldsymbol{x}_\xi) \right\rangle$$

$$\left\langle \mathbf{1}_m, \sigma'(\mathbf{W}\boldsymbol{x}_\alpha) \odot \sigma(\mathbf{W}\boldsymbol{x}_\beta) \odot \sigma'(\mathbf{W}\boldsymbol{x}_\gamma) \odot \sigma(\mathbf{W}\boldsymbol{x}_\xi) \right\rangle,$$

because there is no expressions like $\delta(\cdot)$ and $\delta'(\cdot)$.

We now calculate the expectation of $K_0^{(2)}$, $K_0^{(3)}$ and $K_0^{(4)}$ under Gaussian initialization. First, let us note that $\mathbb{E}K_0^{(3)} = 0$. In fact, since the $r$-th moment of $\mathcal{N}(0,1)$ is zero for odd $r$, we have $\mathbb{E}K^{(r)} = 0$ for any odd $r$.

### B.3.1 Expectation of the second-order kernel

For $K_0^{(2)}$, we need to calculate the following two quantities:

$$\mathbb{E}\left\langle \sigma'(\mathbf{W}\boldsymbol{x}_\alpha) \odot \mathbf{a}_t, \sigma'(\mathbf{W}\boldsymbol{x}_\beta) \odot \mathbf{a}_t \right\rangle, \qquad \mathbb{E}\left\langle \sigma(\mathbf{W}\boldsymbol{x}_\alpha), \sigma(\mathbf{W}\boldsymbol{x}_\beta) \right\rangle.$$

For the first term, we have

$$\mathbb{E}\left\langle \sigma'(\mathbf{W}\boldsymbol{x}_\alpha) \odot \mathbf{a}_t, \sigma'(\mathbf{W}\boldsymbol{x}_\beta) \odot \mathbf{a}_t \right\rangle = m\mathbb{E}_{Z \sim \mathcal{N}(0, I_d)}\left[ \sigma'(\boldsymbol{x}_\alpha^\top Z)\sigma'(\boldsymbol{x}_\beta^\top Z) \right],$$

whereas for the second term, we have

$$\mathbb{E}\left\langle \sigma(\mathbf{W}\boldsymbol{x}_\alpha), \sigma(\mathbf{W}\boldsymbol{x}_\beta) \right\rangle = m\mathbb{E}_{Z \sim \mathcal{N}(0, I_d)}\left[ \sigma(\boldsymbol{x}_\alpha^\top Z)\sigma(\boldsymbol{x}_\beta^\top Z) \right].$$

The above quantities are calculated in many literature (see, e.g., Cho & Saul 2010; Arora et al. 2019a; Bietti & Mairal 2019). Let $\Delta_{\alpha\beta} \in [0, \pi)$ be the angle between $x_\alpha$ and $x_\beta$. We have

$$\mathbb{E}_{Z \sim \mathbb{N}(0, I_d)}\left[ \sigma'(\boldsymbol{x}_\alpha^\top Z)\sigma'(\boldsymbol{x}_\beta^\top Z) \right] = \frac{\pi - \Delta_{\alpha\beta}}{2\pi}$$

$$\mathbb{E}_{Z \sim \mathcal{N}(0, I_d)}\left[ \sigma(\boldsymbol{x}_\alpha^\top Z)\sigma(\boldsymbol{x}_\beta^\top Z) \right] = \frac{\|x_\alpha\|_2 \|x_\beta\|_2}{2\pi} \cdot \left( \sin \Delta_{\alpha\beta} + (\pi - \Delta_{\alpha\beta}) \cos \Delta_{\alpha\beta} \right).$$

Hence, we have

$$\mathbb{E}K_0^{(2)}(\boldsymbol{x}_\alpha, \boldsymbol{x}_\beta) = \boldsymbol{x}_\alpha^\top \boldsymbol{x}_\beta \cdot \frac{\pi - \Delta_{\alpha\beta}}{2\pi} + \|x_\alpha\|_2 \|x_\beta\|_2 \cdot \frac{\sin \Delta_{\alpha\beta} + (\pi - \Delta_{\alpha\beta}) \cos \Delta_{\alpha\beta}}{2\pi}.$$

### B.3.2 Expectation of the fourth-order kernel

We now try to compute $\mathbb{E}K_0^{(4)}$. Inspecting Equation (15), we notice that it suffices to calculate the expectation of the following quantities:

$$\mathrm{I} = \left\langle \mathbf{a}^{\odot 2}, \sigma'(\mathbf{W}\boldsymbol{x}_\alpha) \odot \sigma'(\mathbf{W}\boldsymbol{x}_\beta) \odot \sigma'(\mathbf{W}\boldsymbol{x}_\gamma) \odot \sigma'(\mathbf{W}\boldsymbol{x}_\xi) \right\rangle$$

$$\mathrm{II} = \left\langle \mathbf{1}_m, \sigma'(\mathbf{W}\boldsymbol{x}_\alpha) \odot \sigma'(\mathbf{W}\boldsymbol{x}_\beta) \odot \sigma(\mathbf{W}\boldsymbol{x}_\gamma) \odot \sigma(\mathbf{W}\boldsymbol{x}_\xi) \right\rangle$$

$$\mathrm{III} = \left\langle \mathbf{a}^{\odot 2}, \sigma''(\mathbf{W}\boldsymbol{x}_\alpha) \odot \sigma(\mathbf{W}\boldsymbol{x}_\beta) \odot \sigma'(\mathbf{W}\boldsymbol{x}_\gamma) \odot \sigma'(\mathbf{W}\boldsymbol{x}_\xi) \right\rangle$$

$$\mathrm{IV} = \left\langle \mathbf{a}^{\odot 4}, \sigma''(\mathbf{W}\boldsymbol{x}_\alpha) \odot \sigma''(\mathbf{W}\boldsymbol{x}_\beta) \odot \sigma'(\mathbf{W}\boldsymbol{x}_\gamma) \odot \sigma'(\mathbf{W}\boldsymbol{x}_\xi) \right\rangle$$

$$\mathrm{V} = \left\langle \mathbf{a}^{\odot 4}, \sigma'''(\mathbf{W}\boldsymbol{x}_\alpha) \odot \sigma'(\mathbf{W}\boldsymbol{x}_\beta) \odot \sigma'(\mathbf{W}\boldsymbol{x}_\gamma) \odot \sigma'(\mathbf{W}\boldsymbol{x}_\xi) \right\rangle.$$

The rest of the terms can be calculated similarly.

**The first term.** For the first term, we have

$$\mathbb{E}\left\langle \mathbf{a}^{\odot 2}, \sigma'(\mathbf{W}\boldsymbol{x}_\alpha) \odot \sigma'(\mathbf{W}\boldsymbol{x}_\beta) \odot \sigma'(\mathbf{W}\boldsymbol{x}_\gamma) \odot \sigma'(\mathbf{W}\boldsymbol{x}_\xi) \right\rangle = m\mathbb{P}\left( \boldsymbol{x}_\alpha^\top Z \geq 0, \boldsymbol{x}_\beta^\top Z \geq 0, \boldsymbol{x}_\gamma^\top Z \geq 0, \boldsymbol{x}_\xi^\top Z \geq 0 \right),$$

Note that this term only depends on the angles, so we can WLOG assume that all four vectors lies on the sphere. We reduce this $d$-dimensional integral to a four-dimensional one. For any orthogonal matrix $Q$, we have

$$\mathbb{P}\left( \boldsymbol{x}_\alpha^\top Z \geq 0, \boldsymbol{x}_\beta^\top Z \geq 0, \boldsymbol{x}_\gamma^\top Z \geq 0, \boldsymbol{x}_\xi^\top Z \geq 0 \right) = \mathbb{P}\left( (Q\boldsymbol{x}_\alpha)^\top Z \geq 0, (Q\boldsymbol{x}_\beta)^\top Z \geq 0, (Q\boldsymbol{x}_\gamma)^\top Z \geq 0, (Q\boldsymbol{x}_\xi)^\top Z \geq 0 \right).$$

We choose a $Q$ s.t $(Q\boldsymbol{x}_\alpha)_i = 0$ for $i \geq 2$, $(Q\boldsymbol{x}_\beta)_i = 0$ for $i \geq 3$, $(Q\boldsymbol{x}_\gamma)_i = 0$ for $i \geq 4$, and $(Q\boldsymbol{x}_\xi)_i = 0$ for $i \geq 5$. Moreover, we require $(Q\boldsymbol{x}_\alpha)_1 = 1$. Those requirements specify a unique (up to flips in one direction) rotation matrix $Q$. In order the preserve the angles, we necessarily have

$$Q\boldsymbol{x}_\alpha = \begin{pmatrix} 1 & 0 & 0 & 0 & 0 & \cdots & 0 \end{pmatrix}^\top$$

$$(Q\boldsymbol{x}_\beta)^\top (Q\boldsymbol{x}_\alpha) = \cos \Delta_{\alpha\beta}, \quad \|Q\boldsymbol{x}_\beta\|_2 = 1$$

$$(Q\boldsymbol{x}_\gamma)^\top (Q\boldsymbol{x}_\alpha) = \cos \Delta_{\alpha\gamma}, \quad (Q\boldsymbol{x}_\gamma)^\top (Q\boldsymbol{x}_\beta) = \cos \Delta_{\beta\gamma}, \quad \|Q\boldsymbol{x}_\gamma\|_2 = 1$$

$$(Q\boldsymbol{x}_\xi)^\top (Q\boldsymbol{x}_\alpha) = \cos \Delta_{\alpha\xi}, \quad (Q\boldsymbol{x}_\xi)^\top (Q\boldsymbol{x}_\beta) = \cos \Delta_{\beta\xi}, \quad (Q\boldsymbol{x}_\xi)^\top (Q\boldsymbol{x}_\gamma) = \cos \Delta_{\gamma\xi}, \quad \|Q\boldsymbol{x}_\xi\|_2 = 1.$$

Solving the above system of equations gives

$$(Q\boldsymbol{x}_\beta) = [\cos \Delta_{\alpha\beta}, \sin \Delta_{\alpha\beta}, 0, \cdots, 0]^\top$$

$$(Q\boldsymbol{x}_\gamma) = \left[ \cos \Delta_{\alpha\gamma}, (\sin^{-1} \Delta_{\alpha\beta}) \cdot (\cos \Delta_{\beta\gamma} - \cos \Delta_{\alpha\beta} \cos \Delta_{\alpha\gamma}), \sqrt{\sin^2 \Delta_{\alpha\gamma} - (Q\boldsymbol{x}_\gamma)_2^2}, 0, \cdots, 0 \right)^\top$$

$$(Q\boldsymbol{x}_\xi) = \left[ \cos \Delta_{\alpha\xi}, (\sin^{-1} \Delta_{\alpha\beta}) \cdot (\cos \Delta_{\beta\xi} - \cos \Delta_{\alpha\beta} \cos \Delta_{\alpha\xi}), \frac{\cos \Delta_{\gamma\xi} - (Q\boldsymbol{x}_\gamma)_2 (Q\boldsymbol{x}_\xi)_2}{(Q\boldsymbol{x}_\gamma)_3}, \right.$$
$$\left. \sqrt{\sin^2 \Delta_{\alpha\xi} - (Q\boldsymbol{x}_\xi)_2^2 - (Q\boldsymbol{x}_\xi)_3^2}, 0, \cdots, 0 \right]^\top.$$

Let $v_\alpha, v_\beta, v_\gamma, v_\xi$ be the vectors composed of the first four coordinates of $Q\boldsymbol{x}_\alpha, Q\boldsymbol{x}_\beta, Q\boldsymbol{x}_\gamma, Q\boldsymbol{x}_\xi$, respectively. Then we have

$$\mathbb{P}\left( \boldsymbol{x}_\alpha^\top Z \geq 0, \boldsymbol{x}_\beta^\top Z \geq 0, \boldsymbol{x}_\gamma^\top Z \geq 0, \boldsymbol{x}_\xi^\top Z \geq 0 \right) = \mathbb{P}\left( v_\alpha^\top Z \geq 0, v_\beta^\top Z \geq 0, v_\gamma^\top Z \geq 0, v_\xi^\top Z \geq 0 \right),$$

where with a slight abuse of notation, the vector $Z \sim \mathcal{N}(0, I_4)$ is now a four-dimensional standard Gaussian. Let

$$V = V_{\alpha,\beta,\gamma,\xi} = \begin{pmatrix} v_\alpha^\top \\ v_\beta^\top \\ v_\gamma^\top \\ v_\xi^\top \end{pmatrix} \in \mathbb{R}^{4\times4}. \tag{17}$$

Then the quantity of interest becomes $\mathbb{P}(VZ \in \mathcal{H})$, where $\mathcal{H}$ is the positive orthant of $\mathbb{R}^4$. It's clear from the construction that $V$ is invertible, so we are interested in $\mathbb{P}(Z \in V^{-1}\mathcal{H})$. The set $V^{-1}\mathcal{H}$ is the *positive hull* made by the four columns of $V^{-1}$. Since the law of $Z$ is spherically symmetric, the measure of $V^{-1}H$ under the law of $Z$ is the fraction of the unit sphere $\mathbb{S}^3$ in this hull, which is equal to $\Omega/(2\pi^2)$, where $2\pi^2$ is the surface area of $\mathbb{S}^3$, and $\Omega$ is the *solid angle* for the four column vectors of $V^{-1}$. The solid angle for $r$ vectors has analytical formulas if $r \leq 3$. For $r = 4$ and higher dimensional, there is a formula in terms of multivariate Taylor series (see, e.g., Aomoto 1977; Ribando 2006), but no closed-form formulas are known to the best of our knowledge. However, the probability we are interested in can be efficiently simulated by law of large numbers, because we have reduce the $d$-dimensional Gaussian integral to a four-dimensional one.

In summary, we have the following expression:

$$\mathbb{E}I = m\mathbb{P}(VZ \succeq 0), \qquad Z \sim \mathcal{N}(0, I_4).$$

**The second term.** For the second term, we have

$$\mathbb{E}\left\langle \mathbf{1}_m, \sigma'(\mathbf{W}\boldsymbol{x}_\alpha) \odot \sigma'(\mathbf{W}\boldsymbol{x}_\beta) \odot \sigma(\mathbf{W}\boldsymbol{x}_\gamma) \odot \sigma(\mathbf{W}\boldsymbol{x}_\xi) \right\rangle = m\mathbb{E}\left[ x_\gamma^\top Z \cdot x_\xi^\top Z \cdot \mathbb{1}\left\{ x_\alpha^\top Z \geq 0, \boldsymbol{x}_\beta^\top Z \geq 0, \boldsymbol{x}_\gamma^\top Z \geq 0, \boldsymbol{x}_\xi^\top Z \geq 0 \right\} \right].$$

Similarly, we have

$$\mathbb{E}_{Z \sim \mathcal{N}(0, I_d)}\left[ x_\gamma^\top Z \cdot \boldsymbol{x}_\xi^\top Z \cdot \mathbb{1}\left\{ x_\alpha^\top Z \geq 0, \boldsymbol{x}_\beta^\top Z \geq 0, \boldsymbol{x}_\gamma^\top Z \geq 0, \boldsymbol{x}_\xi^\top Z \geq 0 \right\} \right]$$

$$= \|\boldsymbol{x}_\gamma\|_2 \|\boldsymbol{x}_\xi\|_2 \cdot \mathbb{E}_{Z \sim \mathcal{N}(0, I_4)}\left[ v_\gamma^\top Z \cdot v_\xi^\top Z \cdot \mathbb{1}\{V_{\alpha,\beta,\gamma,\xi} Z \in \mathcal{H}\} \right],$$

which gives the following expression:

$$\mathbb{E}II = m\|\boldsymbol{x}_\gamma\|_2 \|\boldsymbol{x}_\xi\|_2 \cdot \mathbb{E}\left[ v_\gamma^\top Z \cdot v_\xi^\top Z \cdot \mathbb{1}\{V_{\alpha,\beta,\gamma,\xi} Z \succeq 0\} \right] \qquad Z \sim \mathcal{N}(0, I_4).$$

**The third term.** For this term, we have

$$\mathbb{E}III = 3m\mathbb{E}\left[ \delta(\boldsymbol{x}_\alpha^\top Z) \cdot \boldsymbol{x}_\beta^\top Z \cdot \mathbb{1}\left\{ x_\beta^\top Z \geq 0, \boldsymbol{x}_\gamma^\top Z \geq 0, \boldsymbol{x}_\xi^\top Z \geq 0 \right\} \right].$$

Using the same rotation trick, we have

$$III = 3m\mathbb{E}\left[ \delta(\|x_\alpha\|_2 v_\alpha^\top Z) \cdot \|x_\beta\|_2 v_\beta^\top Z \cdot \mathbb{1}\left\{ v_\beta^\top Z \geq 0, v_\gamma^\top Z \geq 0, v_\xi^\top Z \geq 0 \right\} \right], \qquad Z \sim \mathcal{N}(0, I_4).$$

Since $v_\alpha = (1, 0, 0, 0)^\top$, we have

$$\mathbb{E}\left[ \delta(\|x_\alpha\|_2 Z_1) \cdot v_\beta^\top Z \cdot \mathbb{1}\left\{ v_\beta^\top Z \geq 0, v_\gamma^\top Z \geq 0, v_\xi^\top Z \geq 0 \right\} \right]$$

$$= \mathbb{E}_{Z_2, Z_3, Z_4}\left[ \int_{-\infty}^{\infty} dz_1 \frac{1}{\sqrt{2\pi}} e^{-z_1^2/2} \delta(\|x_\alpha\|_2 z_1) h(z_1, Z_2, Z_3, Z_4) \right]$$

$$= \mathbb{E}_{Z_2, Z_3, Z_4}\left[ \frac{1}{\|\boldsymbol{x}_\alpha\|_2 \sqrt{2\pi}} h(0, Z_2, Z_3, Z_4) \right]$$

$$= \frac{1}{\|x_\alpha\|_2 \sqrt{2\pi}} \mathbb{E}\left[ v_{\beta,-1}^\top Z_{-1} \cdot \mathbb{1}\left\{ v_{\beta,-1}^\top Z_{-1} \geq 0, v_{\gamma,-1}^\top Z_{-1} \geq 0, v_{\xi,-1}^\top Z_{-1} \geq 0 \right\} \right],$$

where we let $h(Z_1, Z_2, Z_3, Z_4) = v_\beta^\top Z \cdot \mathbb{1}\left\{ v_\beta^\top Z \geq 0, v_\gamma^\top Z \geq 0, v_\xi^\top Z \geq 0 \right\}$, and in the second equality, we used the fact that, for $c \neq 0$,

$$\int_{-\infty}^{\infty} f(\boldsymbol{x})\delta(cx)dx = \int_{-\infty}^{\infty} \frac{1}{c} f(\frac{cx}{c})\delta(cx)dcx = \frac{1}{c} \int_{\text{sign}(u)\cdot(-\infty)}^{\text{sign}(u)\cdot(+\infty)} f(\frac{u}{c})\delta(u)du = \frac{1}{|c|} \int_{-\infty}^{\infty} f(\frac{u}{c})\delta(u)du = \frac{f(0)}{|c|}.$$

Note that the above computation isn't too messy, because we rotate $Z$ to align with $x_\alpha$, and the delta function appears only at the $\alpha$ location. Other terms in $K_t^{(4)}$ with similar structures as III should be handled similarly (i.e., rotate $Z$ to align with the axis where the delta function appears).

In summary, we have

$$\mathbb{E}\text{III} = \frac{3m}{\sqrt{2\pi}} \cdot \frac{\|x_\beta\|_2}{\|x_\alpha\|_2} \cdot \mathbb{E}\left[v_{\beta,-1}^\top Z_{-1} \cdot \mathbb{1}\left\{v_{\beta,-1}^\top Z_{-1} \geq 0, v_{\gamma,-1}^\top Z_{-1} \geq 0, v_{\xi,-1}^\top Z_{-1} \geq 0\right\}\right].$$

**The fourth term.** We have

$$\mathbb{E}\text{IV} = 3m\mathbb{E}\left[\delta(\boldsymbol{x}_\alpha^\top Z)\delta(\boldsymbol{x}_\beta^\top Z)\mathbb{1}\left\{\boldsymbol{x}_\gamma^\top Z \geq 0, \boldsymbol{x}_\xi^\top Z \geq 0\right\}\right]$$

$$= 3m\mathbb{E}\left[\delta(\|x_\alpha\|_2 \cdot Z_1) \cdot \delta(\|x_\beta\|_2 \cdot (v_{\beta,1}Z_1 + v_{\beta,2}Z_2)) \cdot \mathbb{1}\left\{\boldsymbol{x}_\gamma^\top Z \geq 0, \boldsymbol{x}_\xi^\top Z \geq 0\right\}\right]$$

$$= 3m\mathbb{E}_{Z_3,Z_4}\left[\int_{-\infty}^\infty dz_2 \frac{1}{\sqrt{2\pi}}e^{-z_2^2/2} \int_{-\infty}^\infty dz_1 \frac{1}{\sqrt{2\pi}}e^{-z_1^2/2} \cdot \delta(\|x_\alpha\|_2 z_1) \cdot \delta(\|x_\beta\|_2 \cdot (v_{\beta,1}z_1 + v_{\beta,2}z_2))\right.$$

$$\left. \cdot \mathbb{1}\left\{v_{\gamma,1}z_1 + v_{\gamma,2}z_2 + v_{\gamma,3}Z_3 + v_{\gamma,4}Z_4 \geq 0, v_{\xi,1}z_1 + v_{\xi,2}z_2 + v_{\xi,3}Z_3 + v_{\xi,4}Z_4 \geq 0\right\}\right]$$

$$= \frac{3m}{\|x_\alpha\|_2 \cdot 2\pi} \cdot \mathbb{E}_{Z_3,Z_4}\left[\int_{-\infty}^\infty dz_2 e^{-z_2^2/2} \cdot \delta(\|x_\beta\|_2 v_{\beta,2}z_2)\right.$$

$$\left. \cdot \mathbb{1}\left\{v_{\gamma,2}z_2 + v_{\gamma,3}Z_3 + v_{\gamma,4}Z_4 \geq 0, v_{\xi,2}z_2 + v_{\xi,3}Z_3 + v_{\xi,4}Z_4 \geq 0\right\}\right]$$

$$= \frac{3m}{2\pi\|x_\alpha\|_2\|x_\beta\|_2|v_{\beta,2}|}\mathbb{P}(v_{\gamma,3}Z_3 + v_{\gamma,4}Z_4 \geq 0, v_{\xi,3}Z_3 + v_{\xi,4}Z_4 \geq 0).$$

The probability in the RHS can be calculated explicitly. By spherical symmetry of Gaussian, we have

$$\mathbb{P}(v_{\gamma,3}Z_3 + v_{\gamma,4}Z_4 \geq 0, v_{\xi,3}Z_3 + v_{\xi,4}Z_4 \geq 0) = \frac{\pi - \angle(v_{\gamma,3:4}, v_{\xi,3:4})}{2\pi},$$

where $\angle(v_{\gamma,3:4}, v_{\xi,3:4}) = \arccos(\frac{v_{\gamma,3:4}^\top v_{\xi,3:4}}{\|v_{\gamma,3:4}\|_2\|v_{\xi,3:4}\|_2}) \in [0,\pi)$ is the angle between the two vectors $(v_{\gamma,3}, v_{\gamma,4}), (v_{\xi,3}, v_{\xi,4})$. Hence, we arrive at

$$\mathbb{E}\text{IV} = \frac{3m[\pi - \angle(v_{\gamma,3:4}, v_{\xi,3:4})]}{4\pi^2\|x_\alpha\|_2\|x_\beta\|_2|v_{\beta,2}|}.$$

**The fifth term.** We have

$$\mathbb{E}\text{V} = 3m\mathbb{E}\left[\delta'(\boldsymbol{x}_\alpha^\top Z) \cdot \mathbb{1}\left\{x_\beta^\top Z \geq 0, \boldsymbol{x}_\gamma^\top Z \geq 0, \boldsymbol{x}_\xi^\top Z \geq 0\right\}\right]$$

$$= 3m\mathbb{E}\left[\delta'(\|x_\alpha\|_2 Z_1) \cdot \mathbb{1}\left\{v_\beta^\top Z \geq 0, v_\gamma^\top Z \geq 0, v_\xi^\top Z \geq 0\right\}\right]$$

$$= \frac{3m}{\sqrt{2\pi}}\mathbb{E}_{Z_2,Z_3,Z_4}\left[\int_{-\infty}^\infty dz_1 \cdot \delta'(\|x_\alpha\|_2 z_1) \cdot e^{-z_1^2/2}\right.$$

$$\left. \cdot \sigma'(v_{\beta,1}z_1 + v_{\beta,-1}^\top Z_{-1})\sigma'(v_{\gamma,1}z_1 + v_{\gamma,-1}^\top Z_{-1})\sigma'(v_{\xi,1}z_1 + v_{\xi,-1}^\top Z_{-1})\right]$$

$$= \frac{3m}{\|x_\alpha\|_2\sqrt{2\pi}}\mathbb{E}_{Z_1,Z_2,Z_3}\left[\int_{-\infty}^\infty du \cdot \delta'(u) \cdot e^{-u^2/2\|x_\alpha\|_2^2}\right.$$

$$\left. \cdot \sigma'\left(\frac{v_{\beta,1}u}{\|x_\alpha\|_2} + v_{\beta,-1}^\top Z_{-1}\right)\sigma'\left(\frac{v_{\gamma,1}u}{\|x_\alpha\|_2} + v_{\gamma,-1}^\top Z_{-1}\right)\sigma'\left(\frac{v_{\xi,1}u}{\|x_\alpha\|_2} + v_{\xi,-1}^\top Z_{-1}\right)\right]$$

$$= -\frac{3m}{\|x_\alpha\|_2 \sqrt{2\pi}} \mathbb{E}_{Z_2, Z_3, Z_4} \left\{ \frac{d}{du} \left[ e^{-u^2/2\|x_\alpha\|_2^2} \right. \right.$$

$$\left. \left. \cdot \sigma'\left(\frac{v_{\beta,1} u}{\|x_\alpha\|_2} + v_{\beta,-1}^\top Z_{-1}\right) \sigma'\left(\frac{v_{\gamma,1} u}{\|x_\alpha\|_2} + v_{\gamma,-1}^\top Z_{-1}\right) \sigma'\left(\frac{v_{\xi,1} u}{\|x_\alpha\|_2} + v_{\xi,-1}^\top Z_{-1}\right) \right] \bigg|_{u=0} \right\},$$

where the last equality is by integration by part (which essentially defines $\delta(\cdot)$). With some algebra, one can see that the derivative term in the RHS is equal to (with $u = 0$)

$$\frac{v_{\beta,1}}{\|x_\alpha\|_2} \cdot \delta(v_{\beta,-1}^\top Z_{-1}) \sigma'(v_{\gamma,-1}^\top z_{-1}) \sigma'(v_{\xi,-1}^\top Z_{-1})$$

$$+ \frac{v_{\gamma,1}}{\|x_\alpha\|_2} \cdot \sigma'(v_{\beta,-1}^\top Z_{-1}) \delta(v_{\gamma,-1}^\top z_{-1}) \sigma'(v_{\xi,-1}^\top Z_{-1})$$

$$+ \frac{v_{\xi,1}}{\|x_\alpha\|_2} \cdot \sigma'(v_{\beta,-1}^\top Z_{-1}) \sigma'(v_{\gamma,-1}^\top z_{-1}) \delta(v_{\xi,-1}^\top Z_{-1}),$$

For the first term in the above display, we have

$$\mathbb{E}\left[ \delta(v_{\beta,-1}^\top Z_{-1}) \sigma'(v_{\gamma,-1}^\top z_{-1}) \sigma'(v_{\xi,-1}^\top Z_{-1}) \right]$$

$$= \frac{1}{|v_{\beta,2}|\sqrt{2\pi}} \mathbb{E}_{Z_3, Z_4} \left[ \sigma'(v_{\gamma,3:4}^\top Z_{3:4}) \sigma'(v_{\xi,3:4}^\top Z_{3:4}) \right]$$

$$= \frac{\pi - \angle(v_{\gamma,3:4}, v_{\xi,3:4})}{2\pi\sqrt{2\pi} \cdot |v_{\beta,2}|}.$$

Meanwhile, we have

$$\mathbb{E}\left[ \sigma'(v_{\beta,-1}^\top Z_{-1}) \delta(v_{\gamma,-1}^\top z_{-1}) \sigma'(v_{\xi,-1}^\top Z_{-1}) \right]$$

$$= \mathbb{E}_{Z_4}\left[ \frac{1}{2\pi} \iint_{\mathbb{R}^2} e^{-(z_2^2 + z_3^2)/2} \delta(v_{\gamma,2} z_2 + v_{\gamma,3} z_3) \sigma'(v_{\beta,2} z_2) \sigma'(v_{\xi,2} z_2 + v_{\xi,3} z_3 + v_{\xi,4} Z_4) \right].$$

Using the following change of variables:

$$u = v_{\gamma,2} z_2 + v_{\gamma,3} z_3, \qquad w = z_2,$$

so that

$$z_2 = w, \qquad z_3 = \frac{u - v_{\gamma,2} w}{v_{\gamma,3}},$$

we have

$$\mathbb{E}\left[ \sigma'(v_{\beta,-1}^\top Z_{-1}) \delta(v_{\gamma,-1}^\top z_{-1}) \sigma'(v_{\xi,-1}^\top Z_{-1}) \right]$$

$$= \frac{1}{2\pi} \mathbb{E}_{Z_4}\left[ \iint_{\mathbb{R}^2} \frac{1}{|v_{\gamma,3}|} \exp\left\{ -\frac{(w^2 + (v_{\gamma,2} w/v_{\gamma,3})^2)}{2} \right\} \delta(u) \sigma'(v_{\beta,2} w) \sigma'\left(v_{\xi,2} w + v_{\xi,3} \frac{u - v_{\gamma,2} w}{v_{\gamma,3}} + v_{\xi,4} Z_4\right) \right]$$

$$= \frac{1}{2\pi |v_{\gamma,3}|} \mathbb{E}_{Z_4} \int_{-\infty}^{\infty} dw \exp\left\{ -\frac{w^2}{2} \cdot (1 + (v_{\gamma,2}/v_{\gamma,3})^2) \right\} \sigma'(v_{\beta,2} w) \sigma'\left(\left(v_{\xi,2} - \frac{v_{\xi,3} v_{\gamma,2}}{v_{\gamma,3}}\right) w + v_{\xi,4} Z_4\right)$$

$$= \frac{1}{\sqrt{2\pi} \cdot |v_{\gamma,3}|} \mathbb{E}_{Z_4, W}\left[ \sigma'(v_{\beta,2} w) \sigma'\left(\left(v_{\xi,2} - \frac{v_{\xi,3} v_{\gamma,2}}{v_{\gamma,3}}\right) w + v_{\xi,4} Z_4\right) \right],$$

where $W \sim \mathcal{N}(0, \frac{1}{1 + v_{\gamma,2}^2/v_{\gamma,3}^2})$. By rescaling, for $(X, Y) \sim \mathcal{N}(0, I_2)$, we have

$$\mathbb{E}\left[ \sigma'(v_{\beta,-1}^\top Z_{-1}) \delta(v_{\gamma,-1}^\top z_{-1}) \sigma'(v_{\xi,-1}^\top Z_{-1}) \right]$$

$$= \frac{1}{\sqrt{2\pi}|v_{\gamma,3}|} \mathbb{E}_{X,Y}\left[ \sigma'\left(\frac{v_{\beta,2}}{\sqrt{1 + v_{\gamma,2}^2/v_{\gamma,3}^2}} \cdot X\right) \sigma'\left(\frac{v_{\xi,2} v_{\gamma,3} - v_{\xi,3} v_{\gamma,2}}{v_{\gamma,3}\sqrt{1 + v_{\gamma,2}^2/v_{\gamma,3}^2}} X + v_{\xi,4} Y\right) \right]$$

$$= \frac{1}{2\pi\sqrt{2\pi}|v_{\gamma,3}|} \cdot \left[ \pi - \angle\left(\left(\sqrt{1 + v_{\gamma,2}^2/v_{\gamma,3}^2}, 0\right), \left(\frac{v_{\xi,2} v_{\gamma,3} - v_{\xi,3} v_{\gamma,2}}{v_{\gamma,3}\sqrt{1 + v_{\gamma,2}^2/v_{\gamma,3}^2}}, v_{\xi,4}\right)\right) \right].$$

Now let us consider

$$\mathbb{E}\left[\sigma'(v_{\beta,-1}^\top Z_{-1})\sigma'(v_{\gamma,-1}^\top z_{-1})\delta(v_{\xi,-1}^\top Z_{-1})\right]$$

$$= \frac{1}{2\pi\sqrt{2\pi}}\int_{\mathbb{R}^3} e^{-(z_2^2+z_3^2+z_4^2)/2}\delta(v_{\xi,2}z_2 + v_{\xi,3}z_3 + v_{\xi,4}z_4)\sigma'(v_{\beta,2}z_2)\sigma'(v_{\gamma,2}z_2 + v_{\gamma,3}z_3)dz_2 dz_3 dz_4.$$

We use the following change of variables:

$$x = v_{\xi,2}z_2 + v_{\xi,3}z_3 + v_{\xi,4}, \quad y = z_3, \quad z = z_4,$$

so that

$$z_2 = \frac{x - v_{\xi,3}y - v_{\xi,4}z}{v_{\xi,2}}, \quad z_3 = y, \quad z_4 = z.$$

Then we have

$$\mathbb{E}\left[\sigma'(v_{\beta,-1}^\top Z_{-1})\sigma'(v_{\gamma,-1}^\top z_{-1})\delta(v_{\xi,-1}^\top Z_{-1})\right]$$

$$= \frac{1}{2\pi\sqrt{2\pi}|v_{\xi,2}|}\int_{\mathbb{R}^3}\exp\{-\frac{(\boldsymbol{x} - v_{\xi,3}y - v_{\xi,4}z)^2/v_{\xi,2}^2 + y^2 + z^2}{2}\}\cdot\sigma'\left(\frac{v_{\beta,2}(\boldsymbol{x} - v_{\xi,3}y - v_{\xi,4}z)}{v_{\xi,2}}\right)$$

$$\cdot\sigma'\left(\frac{v_{\gamma,2}(\boldsymbol{x} - v_{\xi,3}y - v_{\xi,4}z)}{v_{\xi,2}}\right)\delta(\boldsymbol{x})dxdydz,$$

where $1/|v_{\xi,2}|$ is the Jacobian term when we do change of variables. Let us define $\Sigma_\xi$ via

$$\Sigma_\xi^{-1} = \begin{pmatrix} 1 + v_{\xi,3}^2/v_{\xi,2}^2 & v_{\xi,3}v_{\xi,4}/v_{\xi,2} \\ v_{\xi,3}v_{\xi,4}/v_{\xi,2} & 1 + v_{\xi,4}^2/v_{\xi,2}^2 \end{pmatrix}.$$

Integrating $x$ out, we get

$$\mathbb{E}\left[\sigma'(v_{\beta,-1}^\top Z_{-1})\sigma'(v_{\gamma,-1}^\top z_{-1})\delta(v_{\xi,-1}^\top Z_{-1})\right]$$

$$= \frac{\det(\Sigma_\xi)^{1/2}}{\sqrt{2\pi}|v_{\xi,2}|}\frac{\det(\Sigma_\xi)^{-1/2}}{2\pi}\int_{\mathbb{R}^3}\exp\left\{-(y,z)\Sigma_\xi^{-1}(y,z)^\top/2\right\}\sigma'\left(-\frac{v_{\beta,2}(v_{\xi,3}y + v_{\xi,4}z)}{v_{\xi,2}}\right)$$

$$\cdot\sigma'\left((v_{\gamma,3} - \frac{v_{\gamma,2}v_{\xi,3}}{v_{\xi,2}})y - \frac{v_{\gamma,2}v_{\xi,4}}{v_{\xi,2}}z\right)dydz$$

$$= \frac{\det(\Sigma_\xi)^{1/2}}{\sqrt{2\pi}|v_{\xi,2}|}\mathbb{E}\left[\sigma'\left(-\frac{v_{\beta,2}(v_{\xi,3}Y + v_{\xi,4}Z)}{v_{\xi,2}}\right)\cdot\sigma'\left((v_{\gamma,3} - \frac{v_{\gamma,2}v_{\xi,3}}{v_{\xi,2}})Y - \frac{v_{\gamma,2}v_{\xi,4}}{v_{\xi,2}}Z\right)\right],$$

where $(Y,Z) \sim \mathcal{N}(0,\Sigma_\xi)$. Note that for another independent vector $(\tilde{Y},\tilde{Z}) \sim \mathcal{N}(0,I_2)$, we have

$$(Y,Z) =_d \Sigma_\xi^{1/2}(\tilde{Y},\tilde{Z}).$$

Hence, we have

$$\mathbb{E}\left[\sigma'(v_{\beta,-1}^\top Z_{-1})\sigma'(v_{\gamma,-1}^\top z_{-1})\delta(v_{\xi,-1}^\top Z_{-1})\right]$$

$$= \frac{\det(\Sigma_\xi)^{1/2}}{\sqrt{2\pi}|v_{\xi,2}|}\mathbb{E}\left[\sigma'\left(-\frac{v_{\beta,2}}{v_{\xi,2}}\cdot(v_{\xi,3},v_{\xi,4})\cdot\Sigma_\xi^{1/2}\cdot(\tilde{Y},\tilde{Z})^\top\right)\cdot\sigma'\left((v_{\gamma,3} - \frac{v_{\gamma,2}v_{\xi,3}}{v_{\xi,2}}, -\frac{v_{\gamma,2}v_{\xi,4}}{v_{\xi,2}})\cdot\Sigma_\xi^{1/2}(\tilde{Y},\tilde{Z})^\top\right)\right]$$

$$= \frac{\det(\Sigma_\xi)^{1/2}}{2\pi\sqrt{2\pi}|v_{\xi,2}|}\cdot\left[\pi - \angle\left(-\frac{v_{\beta,2}}{v_{\xi,2}}\cdot\Sigma_\xi^{1/2}\cdot\begin{pmatrix} v_{\xi,3} \\ v_{\xi,4} \end{pmatrix}, \Sigma_\xi^{1/2}\cdot\begin{pmatrix} v_{\gamma,3} - \frac{v_{\gamma,2}v_{\xi,3}}{v_{\xi,2}} \\ -\frac{v_{\gamma,2}v_{\xi,4}}{v_{\xi,2}} \end{pmatrix}\right)\right].$$

In summary, we get

$$\mathbb{E}V = \frac{-3m}{\|x_\alpha\|_2^2\sqrt{2\pi}}\cdot\left\{v_{\beta,1}\cdot T_1 + v_{\gamma,1}\cdot T_2 + v_{\xi,1}T_3\right\},$$

where

$$T_1 = \frac{\pi - \angle(v_{\gamma,3:4}, v_{\xi,3:4})}{2\pi\sqrt{2\pi} \cdot |v_{\beta,2}|}$$

$$T_2 = \frac{1}{2\pi\sqrt{2\pi}|v_{\gamma,3}|} \cdot \left[ \pi - \angle\left( \left( \begin{array}{c} \sqrt{1 + v_{\gamma,2}^2/v_{\gamma,3}^2} \\ 0 \end{array} \right), \left( \begin{array}{c} \frac{v_{\xi,2}v_{\gamma,3} - v_{\xi,3}v_{\gamma,2}}{v_{\gamma,3}\sqrt{1 + v_{\gamma,2}^2/v_{\gamma,3}^2}} \\ v_{\xi,4} \end{array} \right) \right) \right]$$

$$T_3 = \frac{\det(\Sigma_\xi)^{1/2}}{2\pi\sqrt{2\pi}|v_{\xi,2}|} \cdot \left[ \pi - \angle\left( -\frac{v_{\beta,2}}{v_{\xi,2}} \cdot \Sigma_\xi^{1/2} \cdot \left( \begin{array}{c} v_{\xi,3} \\ v_{\xi,4} \end{array} \right), \Sigma_\xi^{1/2} \cdot \left( \begin{array}{c} v_{\gamma,3} - \frac{v_{\gamma,2}v_{\xi,3}}{v_{\xi,2}} \\ -\frac{v_{\gamma,2}v_{\xi,4}}{v_{\xi,2}} \end{array} \right) \right) \right],$$

where

$$\Sigma_\xi = \begin{pmatrix} 1 + v_{\xi,3}^2/v_{\xi,2}^2 & v_{\xi,3}v_{\xi,4}/v_{\xi,2} \\ v_{\xi,3}v_{\xi,4}/v_{\xi,2} & 1 + v_{\xi,4}^2/v_{\xi,2}^2 \end{pmatrix}^{-1}.$$

## C  Connections and Differences to Previous Works on Label-Aware Kernels

We discuss the relation between our proposed kernels and two lines of research on label-aware kernels, namely the Kernel Target Alignment (KTA) and the Information Bottleneck (IB) principle.

**Connections to KTA.** Recall that our higher-order regression based kernel is

$$K^{(\mathrm{HR})}(\boldsymbol{x}, \boldsymbol{x}') := \mathbb{E}_{\mathrm{init}} K_0^{(2)}(\boldsymbol{x}, \boldsymbol{x}') + \lambda \mathcal{Z}(\boldsymbol{x}, \boldsymbol{x}', S),$$

where $\mathcal{Z}(\boldsymbol{x}, \boldsymbol{x}', S)$ is an estimator of (label of $\boldsymbol{x}$) $\times$ (label of $\boldsymbol{x}'$). From a high-level, this can be regarded as a specific way to to align with the "optimal" kernel $yy^\top$, because

$$\langle \mathbf{K}^{(\mathrm{HR})}, yy^\top \rangle = \langle \mathbb{E}_{\mathrm{init}}[\mathbf{K}_0^{(2)}], yy^\top \rangle + \lambda \langle \boldsymbol{\mathcal{Z}}, yy^\top \rangle,$$

and the term $\langle \boldsymbol{\mathcal{Z}}, yy^\top \rangle$ is close to one as $\boldsymbol{\mathcal{Z}}$ estimates $yy^\top$ by construction.

**Relations to IB.** Consider the following the model fitting process: $Y \to X \to T \to \hat{Y}$. That is: 1) The nature generates a label $Y \in \{\pm 1\}$, e.g., a cat; 2) Given the label $Y$, the nature further generates a "raw" feature $X$, e.g., an image of a cat; 3) We try to find a feature map which maps $X$ to $T$; 4) We use $T$ to generate a prediction $\hat{Y}$.

The IB principle gives a way to justify "what kind of $T$ is optimal". More explicitly, it poses the following optimization problem:

$$\min_{\substack{p_{T|X} \\ p_{Y|T} \\ p_T}} I(p_X; p_T) - \beta I(p_T; p_Y),$$

where we let $p_{A|B}$ to be the conditional density of $A$ conditional on $B$. Then the "optimal" feature $T$ is a randomized map which sends a specific realization $X = \boldsymbol{x}$ to a random feature $T \sim p_{T|X=\boldsymbol{x}}$.

Note that in the IB formulation, the optimal feature $T$ *has no explicit dependence on $Y$* — it only depends on $Y$ through $X$. This is in sharp contrast to our formulation, where we allow the feature to have explicit dependence on $Y$.

To further illustrate this point, it has been shown that any optimal solution to the IB optimization problem must satisfy the following set of *self-consistent equations* (Tishby et al., 2000):

$$p_{T|X}(t|x) = \frac{p_T(t)}{Z(x;\beta)} \exp\left\{ -\beta D_{KL}(p_{Y|X} \| p_{Y|T}) \right\}$$

$$p(t) = \int p_{T|X}(t|x) p_X(x) dx$$

$$p_{Y|T}(y|t) = \int p_{Y|X}(y|x) p_{X|T}(x|t) dx.$$

Note that the distribution of the optimal feature $T$ only has dependence on $x$, and has no explicit dependence on $Y$, because $Y$ is marginalized in the KL divergence term.

# D  Experimental Details and Additional Results

## D.1  Details on Figure 1(b) and 1(c)

To experiment with different label systems, we use the MS COCO object detection dataset (Lin et al., 2014) because there can be various objects in the same image. In this experiment, we consider 50 images with both cats and chairs, 50 images with both cats and benches, 50 images with both dogs and chairs, and 50 images with both dogs and benches. An image with cats and chairs will include neither dogs nor benches, and similar for other cases. Therefore, in total, there are 100 images with cats, 100 images with dogs, 100 images with chairs and 100 images with benches.

We then consider the same two-layer neural network as in Sec. 3.2 and plot the relative ratio (Eq. (9)) for $K_t^{(2)}$ as $t$ increases, which gives Fig. 1(b) and Fig. 1(c).

## D.2  The Architecture of CNN

The architecture of the CNN used in Sec. 3.1 is as follows: there are seven convolutional layers with kernel size 3 and padding number 1, followed by a fully-connected layer. The output channels of each convolutional layer are: $16k$, $16k$, $16k$, $32k$, $32k$, $64k$, and $64k$, where by default $k = 1$. But we increase $k$ to 16, 8, 4 to get better CNN performance for multi-class classification with 2000, 5000, and 10000 training examples in Table 2. The strides for each convolutional layer is 1 except the fourth and the last, which are 2.

## D.3  Details on LANTK-HR

**The hyper-parameter $\lambda$.** Based on our experiments, the test accuracy is usually a concave function of $\lambda$. So we simply choose the best value among $\{0.001, 0.01, 0.1, 1.0\}$.

**Choice of $\mathcal{Z}(\boldsymbol{x}, \boldsymbol{x}', \boldsymbol{S})$.** Our first choice of $\mathcal{Z}$ is based on a kernel regression. Specifically, for LANTK-KR-V1, we consider

$$\mathcal{Z}(\boldsymbol{x}, \boldsymbol{x}', S) = \sum_{i,j} y_i y_j \psi(\mathbb{E}_{\text{init}} \mathbf{K}_0^{(2)}(\boldsymbol{x}, \boldsymbol{x}'), \mathbb{E}_{\text{init}} \mathbf{K}_0^{(2)}(\boldsymbol{x}_i, \boldsymbol{x}_j)),$$

and for LANTK-KR-V2, we consider

$$\mathcal{Z}(\boldsymbol{x}, \boldsymbol{x}', S) = \sum_{i,j} ((\mathbb{E}_{\text{init}} \mathbf{K}_0^{(2)})^{-1} y)_i ((\mathbb{E}_{\text{init}} \mathbf{K}_0^{(2)})^{-1} y)_j \psi(\mathbb{E}_{\text{init}} \mathbf{K}_0^{(2)}(\boldsymbol{x}, \boldsymbol{x}'), \mathbb{E}_{\text{init}} \mathbf{K}_0^{(2)}(\boldsymbol{x}_i, \boldsymbol{x}_j)),$$

where

$$\psi(\phi_{ab}, \phi_{ij}) = \frac{B - (\phi_{ab} - \phi_{ij})^2}{n^2 B - \sum_{s,t} (\phi_{ab} - \phi_{st})^2}$$

is a normalized similarity measure (smaller $(\phi_{ab} - \phi_{ij})^2$ indicates larger similarity) and $B$ is a constant which is set to be the largest $(\phi_{max} - \phi_{min})^2$ in the training data. Note that the change from $y$ to $(\mathbb{E}_{\text{init}} \mathbf{K}_0^{(2)})^{-1} y$ in CNTK-V2 is inspired from the second term in $K^{(\text{NTH})}(\boldsymbol{x}, \boldsymbol{x}')$.

Our second choice of $\mathcal{Z}$ is based on a linear regression with Fast-Johnson-Lindenstrauss-Transform (FJLT) (Ailon & Chazelle, 2009) and some hand-engineered features. FJLT is used to reduce the computational cost because there are $O(10^8)$ examples in the pairwise dataset[10]. And the hand-engineered features are as follows:

1. For LANTK-FJLT-V1, we use the following features: $\mathbb{E}_{\text{init}} K_0^{(2)}(\boldsymbol{x}, \boldsymbol{x}')$ (the label-agnostic kernel), $\cos\langle \boldsymbol{x}, \boldsymbol{x}' \rangle$ (cos of the angle), $\boldsymbol{x}^T \boldsymbol{x}'$ (the inner product), $||\boldsymbol{x}||_2 ||\boldsymbol{x}'||_2$ (the product of $L_2$ norms), $||\boldsymbol{x} - \boldsymbol{x}'||_2^2$ (Euclidean distance), $|\boldsymbol{x} - \boldsymbol{x}'|_1^1$ ($L_1$ distance), $\langle \boldsymbol{x}, \boldsymbol{x}' \rangle$ (the angle between two vectors), $\sin\langle \boldsymbol{x}, \boldsymbol{x}' \rangle$ (sin of the angle), $RBF(\boldsymbol{x}, \boldsymbol{x}')$ (RBF distance between two vectors), $\rho(\boldsymbol{x}, \boldsymbol{x}')$ (the Pearson correlation coefficient between two vectors).
2. For LANTK-FJLT-V2, in addition to the ten features used in LANTK-FJLT-V2, we also include the same 10 features based on the top five principle components from principal component analysis (PCA).

## D.4 Details on Training

We use the Adam optimizer (Kingma & Ba, 2015) with learning rates $3e^{-4}$ for 300 epochs for the CNN in Sec. 3.1 and for 500 epochs for the 2-layer neural network in Sec. 3.2. For simplicity, we use the parameter with best test performance during the whole training trajectory.

## Footnotes

[6]Here we implicitly assume this hyperplane crosses the origin. The construction without this assumption is similar.

[7]The formula for nonzero $\mathbf{f}_0$ can be obtained by similar but lengthier calculations. Since this is not the focus of this paper, we omit the details.

[8] The method developed in this section applies to neural nets with biases, but with lengthier calculations.

[9] That is, if an index appears twice, we take the sum over this index.

[10]Our implementation is based on `https://github.com/michaelmathen/FJLT` and `https://github.com/dingluo/fwht`.