[Reviews · NeurIPS 2020]

Review 1

Summary and Contributions: UPDATE: Having read the other reviews and the authors' replies, I stand by my report. I thought that the authors answered all my questions sufficiently, and I already very much supported the acceptance of this paper. Furthermore, I appreciate the authors' reply to the other reviewers that emphasize that this is a theoretically motivated work, so I am not concerned about the size of the empirical improvements. Finally, I agree with the authors that MSE loss and gradient flow are standard in the NTK literature. This paper investigates improvements to the Neural Tangent Kernel (NTK), which describes the evolution of infinite-width neural networks under gradient descent. The NTK itself does not evolve in the strict large-width limit and is solely a function of network architecture. The lack of data dependence make the NTK ``label-agnostic,'' in contrast to real trained neural networks. As a solution, this paper proposes two modifications to the NTK at initialization that incorporate training labels in some way. These ``label-aware'' kernels are shown to have better generalization performance on sample datasets as well as better ``local elasticity'' -- a statistic introduced to compare average kernel values on intra-class examples versus inter-class examples.

Strengths: I think this is a nice and novel contribution to the NTK literature. The technical sections are sound, and the experiments are appropriate to support the thesis of the paper. The discussion at the beginning about label systems and the need for label-aware kernels was also extremely clear and compelling; in particular, I'd like to highlight the dog vs. cat / bench vs. chair example, making it clear the problem with label-agnostic kernels. Also, the LANTK-NTH result in equation (8) was a really nice result in particular, and the simple modification in equation (5) for the LANTK-HR was intuitive and well-motivated.

Weaknesses: I overall thought this was a strong paper, but I do have a few questions relating to the empirical results and their interpretation: In Tables 1-3, it's clear that the LANTKs outperform the NTK without label awareness, but underperform the true neural network. Of course that's to be expected, the LANTK is an approximation meant to better mock up the true behavior. However, the percent improvement always seems rather small compared to the true difference, which I think undermines some of these results. Do the authors think this is because of the truncation, or are there other factors at play? It seems like the true neural network is still learning a lot more, and presumably it would be nice to try and characterize that or understand why. Relatedly, as I mentioned above, I found the discussion of label systems and the dog vs. cat and chair vs. bench example very compelling. Is there not some way to experiment with changing label systems directly and characterize the improvement specifically on changing label systems when comparing NTK and LANTK? More broadly, I was wondering whether the goal of this paper is more to learn how to simulate neural networks with kernels (as much of the discussion and also broader impact section supposes), or is it to try and use theoretical tools to explain the behavior of neural networks? If it's the former, aren't kernels still wildly impractical for large datasets? If it's the latter, then shouldn't more of the focus have more been on the LANTK-NTH (which is at least an approximation of the true neural network behavior) and shouldn't there have been more attempts to analyze that solution? I'm asking this because I think it has bearing on the question of the significance of a work like this to the broader community.

Correctness: As far as I can tell, this paper is broadly correct. I do have a few minor questions: On lines 241-242, the authors speculate that their two proposed kernels would behave similarly, given the fact that they have different performance intra-class vs. inter-class and given that they're both quadratic in the labels. Given just that data, that seems a bit of a stretch for me -- why would we expect the matrix M from equation (5) to necessarily end up similar to the solution to the NTH truncation? Of course, it's entirely reasonable to experiment with the LANTK-HR for computational cost reasons, but that comment struck me as needing more justification. The authors never comment on the fact that the NTK has a variance across different realizations at finite width, in addition to evolving in time. I was wondering if the authors thought about this, since presumably the variance contribution to certain observables would be of the same order as their NTH modification. Of course, this doesn't matter if their interest is solely in developing better kernels, but it does matter if they want to accurately model what real neural networks are doing.

Clarity: For the most part, the exposition is rather clear. Some minor points: I found the Hoeffding Decomposition discussion a little technical, despite the ultimate outcome (eq. 4) being rather simple. If there is a simple way to provide intuition or clarify the presentation here, I think it might be helpful. Above Claim 2.1, I wasn't exactly sure what ``second order kernel'' meant. Similarly, on lines 183-184, the authors say that their kernel ``takes the form of the Hoeffding decomposition truncated at the second level'' without defining what a level is. This is eventually explained after equation (6), but I think having a remark earlier could be helpful, especially since while the order refers to the superscript of the kernel, the level I think is the order minus two? In 4.1, I found the giving of the numbers 98 and 48 without much context to be somewhat arbitrary, though I appreciate the point being made about inter-class vs. intra-class. Is there some way to make this clearer here? Lastly, I was wondering when the authors use the term NTK, do they just mean the mean of the wide-limit kernel at initialization? Or can NTK refer to the random variable whose expectation at finite width gives the mean? And does it only refer to the quantity at init? For instance, I might have said that the infinite width NTK is not label aware, but the trained NTK at finite width is -- but I don't think this is how the authors want to use the terminology?

Relation to Prior Work: The introduction gives an overview of large-width training, providing appropriate citations. It also discusses places where this limit is found to be different from finite-width neural networks, giving many citations. I think from this exposition it's clear how this work differs from the prior work.

Reproducibility: Yes

Additional Feedback: UPDATED: I appreciate the authors answering my questions directly in their reply. I'm slightly confused about one aspect of Claim 2.1 and the remarks below. Even for label-agnostic kernels, can't I have good generalization even under a changing of label systems, assuming the data is linearly separable in the kernel space in both ways? For instance, linear regression without any kernel is like using the identity as the kernel, which is obviously label-agnostic. Presumably the data can be such that it performs well even as the task is changed, assuming that interpolation yields good enough results? (Of course, things will obviously improve when the kernel is label aware.) In section 3.2 when discussing the optimal kernel, in this case the kernel is clearly optimal with respect to a fixed label system. More generally in deep learning, we sometimes think of the learned representation as being useful across multiple tasks. (For instance, it might simply encode the fact that there is a dog or cat and chair or bench in such a way that the feature representation can still be used even if the task changes from dog or cat to chair or bench.) Is it clear what such a kernel could look like? Or is this (learning representations that are useful for different tasks) a property of deep neural networks that cannot be captured by kernels? In section 4.2 under ``Multi-class classification'' the authors say that the results support the claim that label information is important when the meaning of the features depend on the label system. While I agree in principle, I was confused how these experiments relate to the idea of changing the task or label system, i.e here the label system is fixed, it's just that there are many classes. Could the authors elaborate?


Review 2

Summary and Contributions: This paper attempts to put label information into NTK to improve the performance since NTK is label-agnostic. The authors proposed two methods, LANTK-HR and LANTK-NTH. They proved LANTK-NTH is theoretically closer to NN than NTK. They empirically verified that LANTK-HR performs better than NTK. They implied that LANTK-HR is closer to NN by showing LANTK-HR is more elastic than NTK.

Strengths: - The theory about LANTK-NTH being closer to NN is sound. - The experiments about local elasticity is convincing. - The paper is well written and easy to understand. - The paper is clearly relevant to the NeurIPS community.

Weaknesses: - The fundamental differences between NTK and NN are the width, the learning rate, the batch norm, etc. This paper does not provide enough detailed information about the NN used, including the learning rate schedule, whether batch norm and weight decay are used, which makes the goal of the paper not very well defined. - I agree that local elasticity is an important phenomenon in NN, and designing label-aware kernels can help. However, it doesn't convince me that LANTK-HR is getting closer to NN, even the paper mentions some empirical observation saying that LANTK-NTH agrees with the intention of LANTK-HR. - The performance of CNN in the experiments is not as good as I expect. The authors chose some specific all conv CNN, which may be the reason that the accuracy is not as expected. However, to my knowledge, it is easy to get much higher accuracy even you use plain all conv CNN. Given that, the improvement brought by LANTK is not significant.

Correctness: I agree with most of the claims and methods in the paper, except that I think the performance of CNN can be much high, and the empirical improvement of LANTK is not significant.

Clarity: The paper is well written. Minor: - I suggest to use bold y to represent the label vector. - define E_init

Relation to Prior Work: The paper discussed most of the previous related works.

Reproducibility: Yes

Additional Feedback: - I think you can show the training/regression trajectories or use some other metrics to show that LANTK-HR is closer to NN. - Can you provide the details about CNN training in your paper? --- post author response comments ---- I agree that the theory about adopting NTH is sound, and LANTK-NTH is theoretically closer to NN than NTK under the assumptions (in some "lazy training" regime). However, I just checked the proof of Theorem 3.2, which is basically applying the results of Huang and Yau (2019). I think the theoretical contribution of Theorem 3.2 is not so significant, especially for the technique. If this is purely a theoretical paper, it needs a few more steps to meet the NeurIPS standard (since Theorem 3.2 is the only theorem of the paper), so that's why we need to take a closer look at the empirical part. Unfortunately, LANTK-NTH is not practical, so the authors came up with a practical kernel LANTK-HR and attempt to convince us two things: a) LANTK-HR is close to LANTK-NTH both empirically and theoretically; b) LANTK-HR has good performance in practice. For a), I don't think it is convincing at all to say these two are close in concept by indicating both are second-level truncations of Hoeffding decomposition. Experimentally, the authors only provide two numbers in Sec 4.1 (also mentioned by R1), which looks also not convincing at all (they should at least provide how these two numbers will be for NTK and LANTK-HR). For b), the improvement is not significant (mentioned by both R1 and R3). If you compare other related works that try to improve NTK (e.g. Li et al. (2019), where enhanced CNTK can meet the performance of AlexNet), the improvement in this paper is actually very minor. Plus, the baseline performance (CNN) given in the paper is much lower than what people report. (I personally trained plain several layer convnet before. You can easily reach at least 90+ for the whole CIFAR-10 dataset following standard LR schedule, not to mention that the binary classification is much easier than 10-class. BTW, for the binary classification, the authors used all training data and testing data and should be no need to "randomly sample" as written.) Therefore, I keep my score.


Review 3

Summary and Contributions: The paper contributes to the line of work around neural tangent kernel (NTK). A problem clearly pointed out is that the current NTK doesn't take label information into account, a sub-optimal design. The authors propose two label-aware kernels via the Hoeffding Decomposition. Their approach is sound and well motivated and they make theoretical contribution (Thm 3.2), which controls a certain error term encountered in their kernel design. Experimental results support their claims. While the performance improvement is consistent, but the gain isn't particularly big. Overall, the paper makes a nice contribution to the NTK literature. I'm inclined to accept provided some concerns are addressed.

Strengths: Soundness: The paper chose to present its theoretical contribution rigorously with clearly stated assumptions and propositions proved. They also accompany heuristics based arguments in their main paper, which can broaden its audience. Their experimental settings are transparent and codes are provided to the reviewer. While I haven't checked them carefully, briefly checking it increases my confidence. Significance: In my view, the authors contribution adds to practitioners' toolbox when dealing with small set of labelled data. While the performance gain isn't big, this could open a line of work that eventually gives significant improvement. Novelty: While Hoeffding Decomposition is well known, its application to design of label aware NTK is new.

Weaknesses: The biggest weakness I can tell is on one hand, the authors develops their theoretical work for least squares loss (line 29) but some of their experiments (code: cnn_multi.py) use the cross-entroy loss. This inconsistency should be clearly pointed out. On the other hand, the use of least squares loss for classification is odd. The motivation of studying the problem this way should be pointed out. Can this be done with the cross entropy loss? Similarly, THM 3.2 is developed for gradient flow, not gradient descent. While it is natural and sensible to borrow intuition from the continuous analogue, possible discrepancy can occur and should be discussed. Alternatively, one may develop the gradient descent version.

Correctness: Their claims are valid as far as I check. But I have not checked their proofs in the appendix, which is around 18 pages long. The biggest problem, however is outlined above in the weakness section. Even if the logic is right, the problem setting can be opaque.

Clarity: The paper is in general well written. Minor suggestions: line 32: give concrete examples of NTK for K; line 51-52: not clearly written; line 129: explain least squares for classification; line 150: g instead of f? Typo? line 238-239: The use of cross entropy loss for multi-classification should be disclosed.

Relation to Prior Work: The paper surveys prior works well.

Reproducibility: Yes

Additional Feedback: Update: Thanks to the authors for addressing my concerns/misunderstandings. While I maintain my score, I would like to express my concerns on the way the authors justify least squares and the way they claim their contributions. "nearly all of the NTK literature, ..., work with least squares loss", "the label is one-hot ... which is similar to the settings in ...". In my view, previous works were not accepted because they compromise on least squares - just because they did so does not prevent us from either improving upon them, or clarifying this is a compromise for theoretical work. I personally don't know any practitioners using least squares on classification problems, and thus its acceptability in practice is questionable. I hope the authors can improve their expositions to enlarge their audience base. I'm also confused by the way the authors claim their contributions. In one response, they emphasized the practical aspect: "Since our focus is to design kernels that simulate NNs and can indeed be computed in practice". However, 1) their theory is on LANTK-NTH, while the experiments are on LANTK-HR; 2) they developed it for gradient flow instead of (stochastic) gradient descent; 3) they motivated their kernel design via Hoeffding decomposition instead of simply stating it is sensible to build label interaction into kernel design. I have a hard time seeing how most contents of the paper are related to LANTK-HR, if their *focus* is on kernel design to be used in practice. In another response, "While the empirical improvement might not look pronounced, we’d like to emphasize that our work is theoretically intended". This is a misinterpretation of my comments and partially contradicts their prior statements, unless they claim major contributions to both. If the authors' claim major contributions to learning theory, the paper would have been a reject for me. I cannot see sufficiently new concepts or techniques being developed, and there is a mismatch between their theory and experiments. In sum, I'm inclined to accept because the authors raised the LANTK design issue. This is an interesting observation and methodology, but I find their expositions lack of focus and misleading at times. I hope the authors can improve these in the next revision.


Review 4

Summary and Contributions: The paper argues the Neural Tangent Kernels underperforming finite neural networks is, in part, due to their label agnostic nature. A more flexible, label-aware kernel is proposed and shown to generalize better. UPDATE: After reading the authors' response, I'm leaving my score and especially my confidence rating unchanged.

Strengths: The paper is very well written. It is easy to follow and strikes a good balance between readability and maths. The main idea - that being aware of labels is necessary - is clearly put forward, and the rest of the paper follows naturally. The Hoeffdinger decomposition as the basis for the kernel is interesting.

Weaknesses: There are no obvious weaknesses to the paper that I can see.

Correctness: The only thing that caught my eye is that "For simplicity, we use the parameter with the best test set performance during the whole training trajectory."

Clarity: It is very well written.

Relation to Prior Work: There is a related works section, and some discussion in the supplementary material.

Reproducibility: Yes

Additional Feedback:

[Author Response · NeurIPS 2020]

We'd like to thank all the reviewers for their illuminating reviews. We were pleased to see quite a few remarks of
the reviewers highlighting the novelty and strength of our paper, including *a nice and novel contribution to the NTK*
*literature, technical sections are sound, extremely clear and compelling, their approach is sound and well motivated,*
*and is very well written.* Below we address major questions raised by the reviewers and will revise the paper accordingly.

**Response to Reviewer #1. R1.1. The goal of this paper.** Our ultimate goal is to explain the behaviors of NNs using
"NN-simulating" kernels. You are right that LANTK-NTH is a better choice in this direction. However, LANTK-NTH
is impractical (footnote 2), so we instead use LANTK-HR in our experiments, because LANTK-HR is similar to
LANTK-NTH in both concepts (Sec. 3.1) and experiments (Sec. 4.1).

**R1.2. Improvement is small.** The improvement of LANTK-HR is limited mainly because we use quite simple
higher-order regression methods (line 248-253). A natural implementation of LANTK-HR and LANTK-NTH requires
$O(n^4)$ complexity, but our approximation requires only $O(n^2)$. In our experiments, the kernels can improve quite a lot
if the second-order component (i.e., $\mathcal{Z}$ in Eq. 5) is good. However, in practice, it's hard to get a good $\mathcal{Z}$ due to the time
complexity issue. Still, our simple approximation points out that adding label information can improve NTK.

**R1.3. Comparing NTK and LANTK in changing label systems.** Good suggestions! Will do this.

**R1.4. Relations between LANTK-HR&NTH.** You are right that the relation between two LANTKs requires more
justification. However, the two numbers still indicates that the "intention" of two LANTKs are similar: they are both
trying to increase the similarity between two examples if they come from the same class. We'll add more analysis there.

**R1.5. The impact of variance at initialization.** The variance indeed comes into play, but we choose to not consider it
in our construction because (1) it is not clear to us how to incorporate the variance component in a kernel, as it has no
explicitly formula; (2) we did an experiment on fitting kernel regressions using the expected (over the initialization)
NTK and the realized NTK, and we found that the former consistently outperforms the latter.

**R1.6. On Claim 2.1.** We can have good generalization under various label systems, *only if* the data is separable in
*both label systems*. Our claim says that there are examples where it's separable in one label system, but not in the other.

**Response to Reviewer #2. R2.1. Experimental details.** Thanks! We'll provide more details in the revision.

**R2.2. The closeness between LANTK-HR and NNs.** We disagree on this point. Both empirical evidence and theoreti-
cally well-justified design intentions show that LANTK-HR is closer to NN. Particularly, LANTK-HR generalizes better
on moderately large datasets like CIFAR-10; (2) it's more locally elastic. This is because its construction is based on
aligning with the optimal kernel, which is somewhat independent of the training dynamics (compared to LANTK-NTH
which explicitly takes advantage of the training dynamics). The suggestion of trying to prove the trajectory closeness is
super interesting and we'll pursue this in the future work.

**R2.3. The limited performance of CNN.** While the empirical improvement might not look pronounced, we'd like to
emphasize that our work is theoretically intended, which echos Reviewer 3's comment *this could open a line of work*
*that eventually gives significant improvement.* Nevertheless, we use current CNN settings for the following reasons: 1)
We simply use the default architecture in the tools for NTK computation (Novak et al., 2020). 2) We choose a relatively
simple architecture (seven-layer CNNs) mainly because we want to compare it to CNTKs with the same architecture,
but the computation cost of deeper CNTKs is huge. 3) The improvement of LANTK is a little limited mainly because
of our simple higher-order regression methods (see more in **R1.2**).

**R2.4. Use bold y and define** $\mathbb{E}_{\text{init}}$. Thanks! We'll revise the paper according to your suggestion.

**Response to Reviewer #3. R3.1. Loss inconsistency.** The cross-entropy loss in the code is used for *CNN training*, not
for NTK/LANTK. For NTK/LANTK, we use kernel regression (implicitly based on least squares loss). For multi-class
classification, the label is one-hot (10-dimensional) which is similar to the settings in Arora et al., (2019), and we feel
it is acceptable in practice. We choose to work with the least squares loss because: (1) to the best of our knowledge,
nearly all of the NTK literature, which we build our results on, work with least squares loss (2) in principle, we can
establish an alternative NTK theory based on cross entropy loss (e.g., Ji and Telgarsky 2019 is based on logistic loss),
but we won't be able to get closed form formulas for the kernel. Since our focus is to design kernels that simulate NNs
and can indeed be computed in practice, we chose to stick to the least squares loss.

**R3.2. Concerns about the gradient flow.** Similar to the above, the reason we work with continuous dynamics is
because: (1) it's analytically simpler; (2) most NTK literature work with gradient flow. We will discuss this point
explicitly in the next version. Developing the gradient descent version is left for future work.

**Response to Reviewer #4. R4.1. Concerns on test performance for CNN training.** This issue only appears in CNN
training, and if we use, say, the last iterate performance, the relative improvement of LANTKs will only be better.
Nevertheless, we thank the reviewer for pointing this out, and we clearly state this issue in our later version.

[Meta-Review · NeurIPS 2020]

Building on the neural tangent kernel (NTK) concept, which describes what happens when a large neural network is initialized with random weights and starts optimization of the weights, the authors defend the idea that a better approximation should involve a kernel that depends on the labels. They derive two label-aware kernels, by approximating the dynamics of the kernel when optimization starts. They provide experimental results suggesting that the label-aware kernels capture work better than the baseline NTK. This paper generated a long and passionate discussion among reviewers. On the one hand, all agreed that this paper is original and brings up new ideas in our quest to better understand the theory of large neural network. It would clearly be of interest to the NeurIPS community, and is likely to trigger further work in the future. On the other hand, there was less consensus about the technical quality and novelty of the contribution, in the sense that no theory (besides intuition) is given to suggest that the new label-aware kernels are good approximations to the neural network, and the experiments are not very convincing either.